# Entropy-Aware Dynamic KV Cache Sparsification for Autoregressive Image Generation and Editing

**Tong Tong** [* 1]   **Ling Xing** [* 2]   **Linjie Li** [3]   **Rui Yan** [2]   **Zhengyuan Yang** [3]   **LiJuan Wang** [3]   **Alex Jinpeng Wang** [4]

## Abstract

Autoregressive (AR) image generation has recently gained momentum as a scalable alternative to diffusion models, benefiting from unified next-token prediction paradigm and strong instruction following ability. However, AR visual generation must decode excessively long sequences of visual tokens, making inference heavily bottlenecked by the memory footprint and latency of the self-attention KV cache. While KV cache compression is well studied in Large Language Model, its counterparts in AR image generation remain underexplored. The reason is fundamental: visual tokens are highly redundant, and their spatial information density is highly non-uniform. In this work, we introduce SparseAR, a training-free, entropy-aware sparse attention method that is specifically designed for AR image generation and editing. Our key insight is that information-rich regions exhibit higher entropy and require broader attention, while redundant regions show lower entropy and allow aggressive sparsification. Based on this insight, we dynamically identify information-rich regions during decoding and adaptively adjust attention sparsity to reduce KV-cache overhead. SparseAR is plug-and-play and can be readily applied to mainstream AR models. Extensive experiments on four representative AR models across multiple benchmarks demonstrate that SparseAR significantly **improves inference efficiency while maintaining, and often even improving, generation and editing quality**.

## 1. Introduction

Autoregressive (AR) models with the next-token prediction paradigm have achieved remarkable success in large lan-

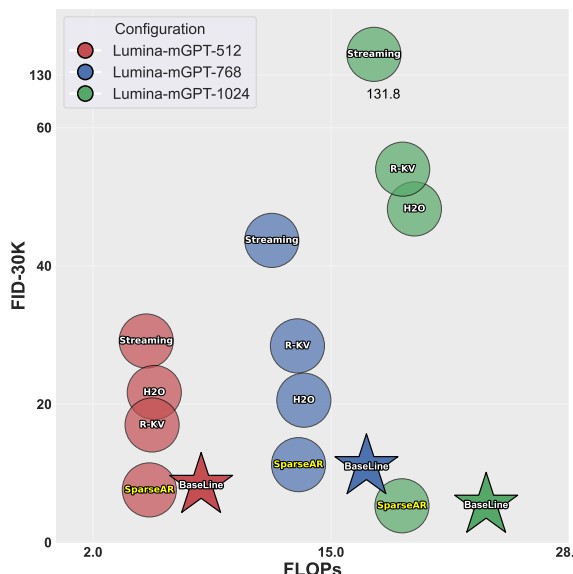

*Figure 1.* **Efficiency *vs*. Generation Quality.** Our SparseAR reduces FLOPs while maintaining generation quality nearly on par with the full-cache baseline (stars), whereas other KV cache compression methods suffer from substantial FID degradation. FID was calculated between 30K generated images and a fixed set of real images, with all methods evaluated under the same compression budget based on Lumina-mGPT.

guage models (LLMs) (Liu et al., 2024a; Vaswani et al., 2017; Achiam et al., 2023) and have also shown excellent performance in visual generation (Liu et al., 2024b; Team, 2024; Xie et al., 2024; Chern et al., 2024). Compared to diffusion models (Chen et al., 2025b; Xin et al., 2025; Wu et al., 2025a), AR models have the potential to become a completely unified model due to their scalability and generality (Xie et al., 2024; Team, 2024; Chen et al., 2025b). However, due to the self-attention mechanism, AR models incur substantial computational overhead and heavy KV cache during inference when processing long sequences (Kong et al., 2025; Bolya et al., 2022). This phenomenon is even more pronounced in tasks like text-to-image generation or image editing, which often require processing extremely large numbers of visual tokens (Chen et al., 2025b; Xin et al., 2025). For instance, generating a $1024 \times 1024$ image with Lumina-mGPT (Liu et al., 2024b)

---

Equal contribution. [1]Beijing University of Posts and Telecommunications [2]Nanjing University of Science and Technology [3]Microsoft [4]Central South University. Correspondence to: Alex Jinpeng Wang <jinpengwang@u.nus.edu>.

*Proceedings of the $43^{rd}$ International Conference on Machine Learning*, Seoul, South Korea. PMLR 306, 2026. Copyright 2026 by the author(s).

requires caching over 4K tokens. In addition, EditAR (Mu et al., 2025) requires processing more than 1K tokens in the pre-fill stage when editing images.

In the Large Language Model (LLM) literature, KV-cache compression and sparse attention have emerged as standard tools to accelerate long-context inference (Xiao et al., 2024b; Zhang et al., 2023; Tang et al., 2024; Wu et al., 2025b; Kong et al., 2025). However, naively applying these works is not the optimal efficiency-performance trade-off. A key reason is that visual tokens exhibit fundamentally different properties: they are highly redundant, and their information distribution is *spatially non-uniform*. Consequently, directly transferring fixed-pattern or modality-agnostic KV sparsification strategies may either prune away crucial details (hurting generation quality) or waste resources on redundant regions (limiting practical gains).

In this work, we revisit KV-Cache efficiency from the perspective of *visual information density*. We observe that, during AR image decoding, different spatial regions demand different levels of long-range dependency: information-rich areas such as faces, text, and fine structures often require broader attention, while large homogeneous regions can tolerate much more aggressive sparsification. Motivated by this insight, we propose SparseAR, a training-free, entropy-aware dynamic sparse attention method tailored for AR image generation and editing. SparseAR leverages the predictive entropy of model as a simple yet effective signal to identify where attention should remain dense and where it can be pruned, dynamically allocating attention budget and KV cache loading throughout decoding. Based on this advantage, SparseAR clearly outperforms the prior KV-cache compression methods (as shown in Figure 1).

Notably, our SparseAR is plug-and-play and suitable for different resolution configurations. We test our method efficiency on four representative unify image generation and editing models, including fully autoregressive models LlamaGen (Sun et al., 2024), Lumina-mGPT (Liu et al., 2024b), and EditAR (Mu et al., 2025), as well as the hybrid AR-diffusion model BLIP3o-Next (Chen et al., 2025a). Extensive experiments across multiple popular benchmarks show our method not only improves inference efficiency but also maintains, and in some cases even improves, image generation and editing quality.

Our contributions are summarized as follows: *i.* We identify a key discrepancy between language and visual autoregressive decoding: visual tokens are spatially redundant and information density is highly non-uniform, making fixed-pattern KV compression suboptimal for image generation and editing. *ii.* We propose SparseAR, a training-free entropy-aware sparse attention method that dynamically allocates attention and KV cache loading based on predictive uncertainty, enabling efficient decoding without restricting

generation dynamics. *iii.* Extensive experiments across four popular AR generators show that SparseAR delivers substantial efficiency gains while preserving, and often improving, generation and editing quality.

## 2. Related Work

**Visual Generation in Autoregressive Models.** Visual autoregressive (AR) models formulate image generation as a next-token prediction problem and have attracted increasing attention for their generality. Early pioneers like Pixel-RNN (So et al., 2024) and PixelCNN (Van den Oord et al., 2016) adopt pixel-level autoregressive generation in a raster-scan order. However, operating in the high-dimensional pixel space resulted in prohibitive computational costs, limiting scalability to high-resolution images. VQVAE (Van Den Oord et al., 2017b) and VQGAN (Yu et al., 2021) address this issue by introducing vector quantization, enabling AR generation in a compact discrete token space. Building on this latent-space paradigm, recent works such as LlamaGen (Sun et al., 2024), Chameleon (Team, 2024), Emu3 (Wang et al., 2024b), and Lumina-mGPT (Liu et al., 2024b) have successfully scaled up AR models, demonstrating capabilities that rival state-of-the-art diffusion models. Nevertheless, autoregressive generation remains bottlenecked by high inference latency, particularly for complex and high-resolution scenes.

**Efficient Visual Generation.** Improving the efficiency of visual generation has attracted growing attention. Diffusion-based methods (Zhu et al., 2024a; Selvaraju et al., 2024; Zou et al., 2024; Shen et al., 2025; Lou et al., 2024; Ma et al., 2024a; Tian et al., 2025; Whalen et al., 2025; Dong et al., 2025) mainly accelerate training and sampling via scheduler optimization, model pruning, and distillation. For AR models, SJD (Teng et al., 2024) and its follow-up variants (Jang et al., 2024; So et al., 2025b; Chen et al., 2025e; Lu et al., 2025; So et al., 2025a) constitute a prominent class of acceleration methods. They leverage a lightweight draft model to propose multiple candidate tokens, which are then validated in parallel by a powerful target model. However, these methods sometimes require additional external training and bear the memory burden of the draft model. Alternatively, some works (Zhu et al., 2024a; Wang et al., 2025b; Ye et al., 2025a; He et al., 2025; Zhang et al., 2025; Pang et al., 2025; Ma et al., 2025a) accelerate AR decoding by generating multiple tokens within a single step, but achieving this requires training in a specific order and pattern. A separate line of research (Li et al., 2025; Qin et al., 2025b; Guo et al., 2025a; Chen et al., 2025d; Ma et al., 2024b; Guo et al., 2025b; Yang et al., 2025; Ren et al., 2025; Kumbong et al., 2025) explores token skipping, pruning, or hierarchical masking to improve efficiency of image generation. These techniques are typically designed

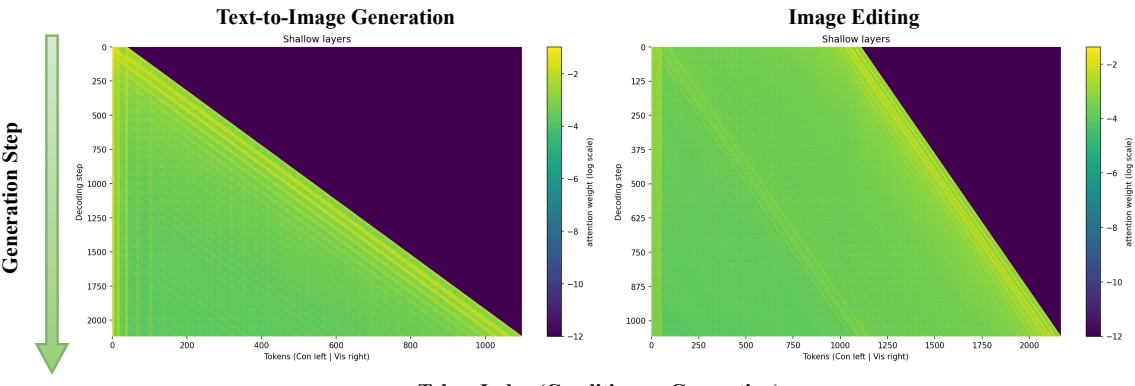

*Figure 2.* **Attention extremely concentrates on local and condition tokens during autoregressive generation.** Visualization of attention allocation in text-to-image generation (left) and image editing (right). Across both settings, the model consistently attends to the conditioning context and nearby visual tokens, while attention to distant visual tokens becomes increasingly sparse as decoding proceeds, revealing substantial redundancy in long-range visual context. The yellow highlighted areas indicate that they receive more attention.

for specific AR paradigms, such as VAR (Tian et al., 2024) or MAR (Li et al., 2024a), and do not readily extend to general AR models.

**Long Context Compression.** Long context compression is a long-standing challenge for LLMs (Kong et al., 2025; Bolya et al., 2022; Xing et al., 2025a;b; Wang et al., 2024a). Representative solutions include KV cache compression (Xiao et al., 2024b; Gu et al., 2025; Xiao et al., 2024a; WEI et al., 2025; Wu et al., 2025b) and Prompt compression (Li et al., 2024b; Jiang et al., 2024). Stream-LLM (Xiao et al., 2024b) enables unbounded text generation by combining sliding-window attention with attention sink tokens, selectively preserving KV states during decoding. Further works (Lu et al., 2024; Jiang et al., 2024) introduce chunk-wise processing and semantic-aware compression. More recent efforts, such as ChunkKV (Liu et al., 2025), ParallelComp (Xiong et al., 2025b), and UNComp (Xiong et al., 2025a) further explore parallelization and uncertainty-aware mechanisms for adaptive KV compression. However, directly extending these techniques to AR image generation is non-trivial, as visual tokens exhibit highly redundant yet spatially non-uniform information distributions, which are not adequately handled by existing text-oriented KV compression methods.

## 3. Methodolgy

### 3.1. Motivation

**Inference Cost in Autoregressive Image Generation.** Similar to LLMs, autoregressive image generation consists of two stages: pre-filling and decoding. During pre-filling, textual prompts or category labels are encoded into $C$ conditioning tokens, initializing the key-value states $\{\mathbf{K}_{\text{cond}}, \mathbf{V}_{\text{cond}}\} \in \mathbb{R}^{C \times D}$. In decoding stage, the model will

take the last generated token to calculate its $K, Q, V$. The model uses $Q$ to multiply with every $K$ of previous tokens to generate the *attention weights*. The attention weights will then get normalized using softmax, where each value $a_i$ represents the attention score between $i$th token and the current token. Besides the time spent on sampling each token, another important factor is that during the decoding phase, the K and V values of existing tokens must be loaded to perform self-attention (Tang et al., 2024; Kwon et al., 2023). This phenomenon is particularly noticeable in image-to-image generation tasks, *e.g.,* image editing, where the input image often requires a large number of tokens for representation, occupying the KV cache for an extended period and slowing down the loading and computation of attention.

**Condition and Local Dependency for Visual Generation.** Previous research (Xiang & Fan, 2025; Xiao et al., 2024b; Zhang et al., 2023) has revealed a significant sparse attention phenomenon in LLMs, indicating that only a small fraction of tokens are needed to contribute most of the attention during the generation process. This situation also exists in visual autoregressive models, as shown in the Figure 2, in text-to-image and image editing, the conditional context and local tokens closer to the generation position often maintain high attention. As the generation process progresses, image tokens further away receive sparse and varying degrees of attention. In particular, the vast majority of tokens contribute very little to attention. This localized attention distribution reveals the redundancy of distant visual tokens, making it intuitive and feasible to progressively discard outdated tokens during the generation process.

**Simple Compression Loss Details.** Based on these observations, is it possible to discard distant visual tokens in the context according to a fixed pattern without affecting generation? To investigate this phenomenon, we conducted

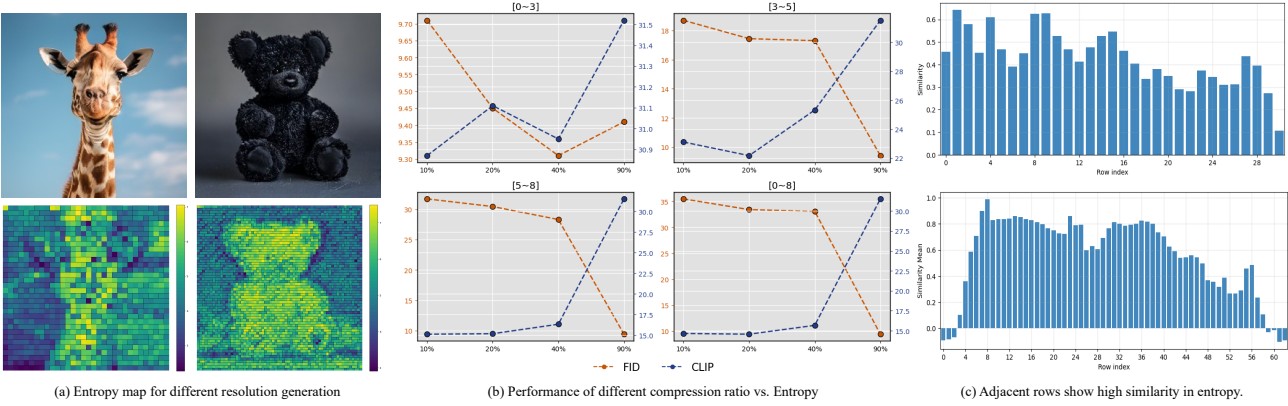

(a) Entropy map for different resolution generation  (b) Performance of different compression ratio vs. Entropy  (c) Adjacent rows show high similarity in entropy.

*Figure 3.* (a). **Next-token prediction entropy can reflect information density.** Information redundancy often exhibits a continuous low entropy distribution, allowing for the use of more restricted contexts in these regions. (b). **Performance of Lumina-mGPT at different compression rates.** The x-axis percentages denote KV retention ratio. By compressing regions with different entropy values to varying degrees, it can be seen that significant context compression in low-entropy regions has a negligible impact on performance. (c). **Entropy similarity distribution of adjacent rows** at 512x512 and 1024x1024 resolutions for Lumina-mGPT.

preliminary experiments. As discovered in Figure 2, the attention in AR image generation often exhibits a pattern of clustering around condition and local tokens. Therefore, we simply design a KV compression scheme: after generating a certain number of tokens, we retain only the text conditions, initial tokens, and most recently generated tokens as KV to guide generation. The specific results are shown in Figure 4, where the generated image in the compressed KV mode has fewer texture details compared to the original model, such as the lion's mane and the shape of the coastline. This confirms that if a position- and pattern-based compression method is used without considering the image content, it will significantly affect the generation result.

**Information Differs between Different Regions.** Unlike discrete text tokens, images exhibit lower information density, significant redundancy and highly non-uniform spatial information distribution (So et al., 2025b; Ma et al., 2025a; Xiang & Fan, 2025). This is mainly due to the subtle high-frequency details in visual tokens and the complex and varied regional patterns in images (Van Den Oord et al., 2017a; Zhu et al., 2024b; Guo et al., 2025b). For example, flat background regions such as the sky tend to contain many tokens of similar colors, leading to redundancy. In complex regions, similar token features can also result from structural consistency. **Therefore, we wonder if the influence of constrained context vary in different image regions.**

**Entropy as Image Information Indicator.** Inspired by recent research ( (Ma et al., 2025b; Pagnoni et al., 2025; Wang et al., 2025a)), we introduce an efficient metric that dynamically indicates model uncertainty. The entropy of the predicted logit is used as a dynamic identifier of the region where generation occurs during the generation process. The meaning of entropy refers to the uncertainty of the predicted

A majestic lion standing on a rocky cliff at golden hour, dramatic clouds.

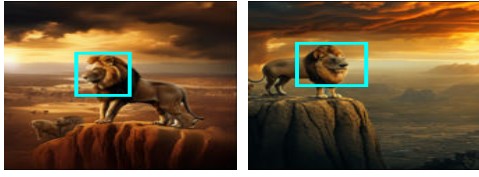

A tranquil sunrise over the ocean with pastel clouds and reflective tidal pools in the foreground.

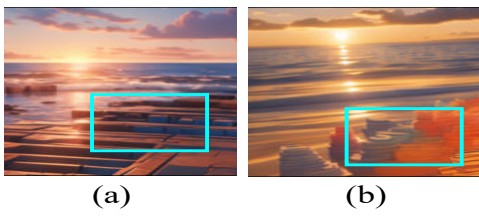

**(a)**  **(b)**

*Figure 4.* (a) **The generated result of LlamaGen-XL without modification.** (b) **The generated result of LlamaGen after applying with a simple compression rule.** After simple, fixed compression, the AR image generation model suffers significant loss of detail. For example, the mane and fur texture of the lions in the first row show significant degradation, and the shape of the reefs in the second row (b) is severely distorted.

logits from model, the entropy $\epsilon$ of log-likelihood over all $V$ codebook entries at each generation step is calculated as follows:

$$\epsilon = -\sum_{k=1}^{V} p_k \log(p_k). \tag{1}$$

As illustrated in Figure 3 (a), areas with simpler content (such as uniform colors) generally demonstrate lower entropy values, whereas highly variable regions containing more complex foreground elements (including objects, structures, and textures) exhibit higher entropy values. Utilizing

this property, we conducted further experimental analysis. We selected 3,000 samples for evaluation on Lumina-mGPT-512 model. As shown in Figure 3 (b), to verify our viewpoint, we directly used the original model to perform next token prediction to get entropy at each step. Then, we performed different degrees of context compression on different generation regions for different entropy value ranges. As can be seen, attention compression within the entropy range of 0 to 3 has almost no impact on generation performance. However, when we apply different degrees of compression to all other regions, a significant decrease in generation performance can be observed.

### 3.2. Adaptive Sparse Attention for Generation

Based on these findings, this paper proposes a simple yet effective KV compression method for autoregressive visual generation. Specifically, we dynamically estimate whether the generation step is located in a flat region with information redundancy or a complex, highly variable region based on the predicted entropy value. One of the most accurate ways to use entropy is to leverage a powerful pre-trained autoregressive model. However, this leads to a significant increase in computational cost. An intuitive understanding is that information in images is clustered. Local regions often exhibit high correlation. We further conducted experimental analysis, calculating entropy values for 3K samples at 512 and 1024 resolutions using Lumina-mGPT (Liu et al., 2024b) and recording the numerical similarity between adjacent positions and adjacent rows. As shown in the Figure 3 (c), the entropy values exhibit a high degree of similarity across most row and position indices, especially in the region from the middle to the end. Therefore, we designed a method based on historical entropy and logits caching to estimate the entropy value of the current region.

**Hyperparameter and Symbolic Formula.** During the autoregressive generation process, the KV cache at step $S_i$ consists of accumulated key-value pairs from both conditioning and previously generated visual tokens, denoted as $\mathbf{P}_{1:i} = \{\mathbf{K}_{1:i}, \mathbf{V}_{1:i}\}$. We partition these historical KV cache into three distinct categories based on their temporal and spatial relevance to the current generation step:

The initial prefix KV pairs $\mathbf{KV}_{\text{prefix}}$ comprise the earliest generated KV pairs, typically including conditioning KV pairs and the first few visual token KV pairs that establish the global style and semantic context. We denote the number of such KV pairs as $P$. The recent local KV pairs $\mathbf{KV}_{\text{local}}$ consist of the most recently generated KV pairs within a small spatial-temporal neighborhood of the current generation position, capturing immediate contextual dependencies. We denote their number as $L$. All remaining historical KV pairs, excluding $\mathbf{KV}_{\text{prefix}}$ and $\mathbf{KV}_{\text{local}}$, are classified as distant KV pairs $\mathbf{KV}_{\text{distant}}$. We denote their number as $S$.

---

**Algorithm 1 Entropy-Aware KV Allocation**

---

1: **Require:** thresholds $(\theta_{\text{low}}, \theta_{\text{high}})$, budgets $(P, L, S)$, activation step $\tau$, current generation step i
2: **for** $i = 1$ **to** $N$ **do**
3:    **if** $i < \tau$ **then**
4:       $\mathbf{KV} \leftarrow \mathbf{P}_{1:i}$
5:    **else**
6:       $\mathcal{E}_i \leftarrow \{e_j : j \in T_i(w) \cup S_i(r)\}$,
7:       $s_i \leftarrow \text{median}(\mathcal{E}_i)$
8:       **Select KV**:
9:       **if** $s_i \geq \theta_{\text{high}}$ **then**
10:          $\mathbf{KV} \leftarrow \mathbf{P}_{1:i}$
11:       **else if** $s_i \leq \theta_{\text{low}}$ **then**
12:          $\mathbf{KV} \leftarrow \mathbf{KV}_{\text{prefix}}(P) \cup \mathbf{KV}_{\text{local}}(L)$
13:       **else**
14:          $\mathbf{KV} \leftarrow \mathbf{KV}_{\text{prefix}}(P) \cup \mathbf{KV}_{\text{local}}(L) \cup \mathbf{KV}_{\text{distant}}^{\text{topk}}(S)$
15:       **end if**
16:    **end if**
17:    Compute logits $\rightarrow e_i$, update $D$, append KV to cache
18: **end for**

---

Therefore, the total KV cache budget for attention computation at each step is $B = P + L + S$. Furthermore, we use $\tau$ to represent the step at which the proposed KV compression method is activated. The image generation process is considered as an $H \times W$ generation process. First, for each generation step $S_i$, we take the logits to calculate the corresponding entropy $e_i$. Simultaneously, we maintain a dictionary $D = \{(i, e_i)\}$ to store these entropy values and record their corresponding positions $i$. Then, Each index $i$ is mapped to 2D coordinates $(r(i), c(i))$. Formally, let the total number of visual tokens be $N = H \times W$.

**Entropy-Aware KV Allocation.** For low-entropy regions, these regions are mostly flat, simple backgrounds, colors, and other highly localized content. We assume that during generation in these regions, only KV which contains global style and local textures (Xiang & Fan, 2025) need to be provided for attention calculation to ensure the generation stability. On the other hand, for high-entropy regions, which typically contain more complex and detailed texture information, we do not prune the KV cache. The attention focused on these regions is often more dispersed, requiring more historical tokens as context. Based on the entropy value, we categorize the current step into high-entropy, low-entropy, and normal regions. To decide what regions the current generation step belongs to, we collect causal temporal $T_i\{w\}$ and $S_i\{r\}$ spatial neighbors that are already generated. Here, $w$ denotes the temporal window size and $r$ denotes the spatial radius used to form the neighbor sets. Next, we estimate the entropy value of the current step based on the median of these local regions and further allocate KV cache based on the classification results. The specific

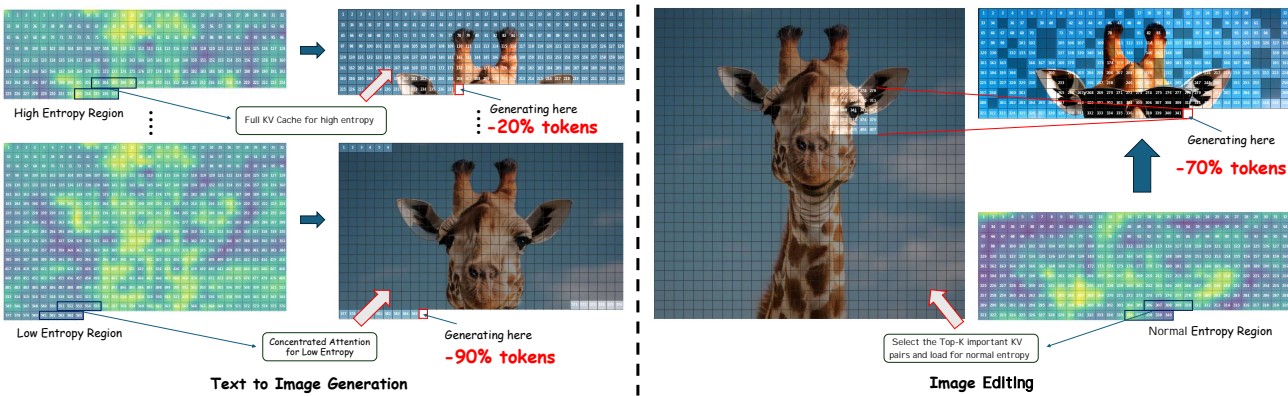

*Figure 5.* **The framework of proposed SparseAR**. For text-to-image generation, SparseAR adaptively distinguishes regions with different entropy levels and applies region-specific sparse attention. For high-entropy regions, SparseAR queries the importance of historical terms and selectively retains all or discards a portion of the KV cache. For low-entropy regions, we force it to focus attention on local and global tokens. For image editing, we observe a strong reliance on conditional attention at every generation step. Accordingly, we dynamically select only region-specific conditional tokens via sparse attention and avoid attention computation on the rest.

process is shown in the algorithm 1. And we provide a more detailed explanation in section A in the appendix A.

### 3.3. Autoregressive Image Generation under SparseAR

Based on the above design and observations, we dynamically apply KV Cache sparsification to the image generation or image editing process. As shown in Figure 5, in the text-to-image generation, we dynamically allocate the KV Cache required for the corresponding region when the AR model generates tokens one by one. By allocating a very small amount of KV Cache to deterministic, homogeneous redundant or repetitive regions as the image context, SparseAR significantly reduces the massive computational cost required for generation.

For the image editing task of the AR model, we adopt a similar KV sparsification method. Notably, when the model performs image editing, the source image is also encoded as a series of conditional tokens and stored in the KV Cache. Unlike text-to-image generation, these conditional token sequences introduce greater contextual information. Therefore, during editing, SparseAR sparsifies not only the KV cache accumulated from generated tokens, but also the condition-image KV cache, retaining only the spatially relevant condition tokens for the current generation step.

## 4. Experiment

### 4.1. Setup

**Models** For evaluating text-to-image generation, we integrate our method with the state-of-the-art autoregressive visual generation model LlamaGen (Sun et al., 2024), Janus-Pro (Chen et al., 2025c), Lumina-mGPT (Liu et al., 2024b). For image editing tasks, three classic models EditAR (Mu

et al., 2025), Lumina-mGPT-Omni (Liu et al., 2024b) and BLIP3o-Next (Chen et al., 2025b) were selected to test image editing task.

**Implementation Details** For other existing KV compression methods used for comparison, we use $B$ to denote the compression budget size for evaluation. For the proposed method SparseAR, we use the generation step $tau$ for both text-to-image generation and image editing. Further discussion of this setting can be found in section 4.4 and the appendix A. For other existing KV compression methods used for comparison, we use $B$ to denote the budget size reported in their respective papers, as shown in section B

**Evaluation Benchmark** For text-to-image generation, we adopt a widely used benchmarks GenEval (Ghosh et al., 2023) to assess high-level semantic alignment and compositional consistency for text-to-image task. We further evaluate image quality and prompt adherence using Frechet Inception Distance (FID) Score and CLIP Score on 30k Laion-COCO (LAION e.V., 2023) validation samples. We denote FID Score as FID-30k. For image editing, we evaluate the effectiveness of our proposed method on image editing tasks using the ImgEdit benchmark (Ye et al., 2025b).

### 4.2. Main Result

**Text-to-Image Generation.** To evaluate the performance of the proposed method on the text-to-image task, we conducted a quantitative evaluation. Notably, we used the number of steps in the generation process that enable SparseAR as a hyperparameter $\tau$. We assume that in AR image generation, the first generated parts establish the overall color and style (Xiang & Fan, 2025; Guo et al., 2025b; Pang et al., 2025), and as more content is generated, the contextual information becomes richer, making the content to be gener-

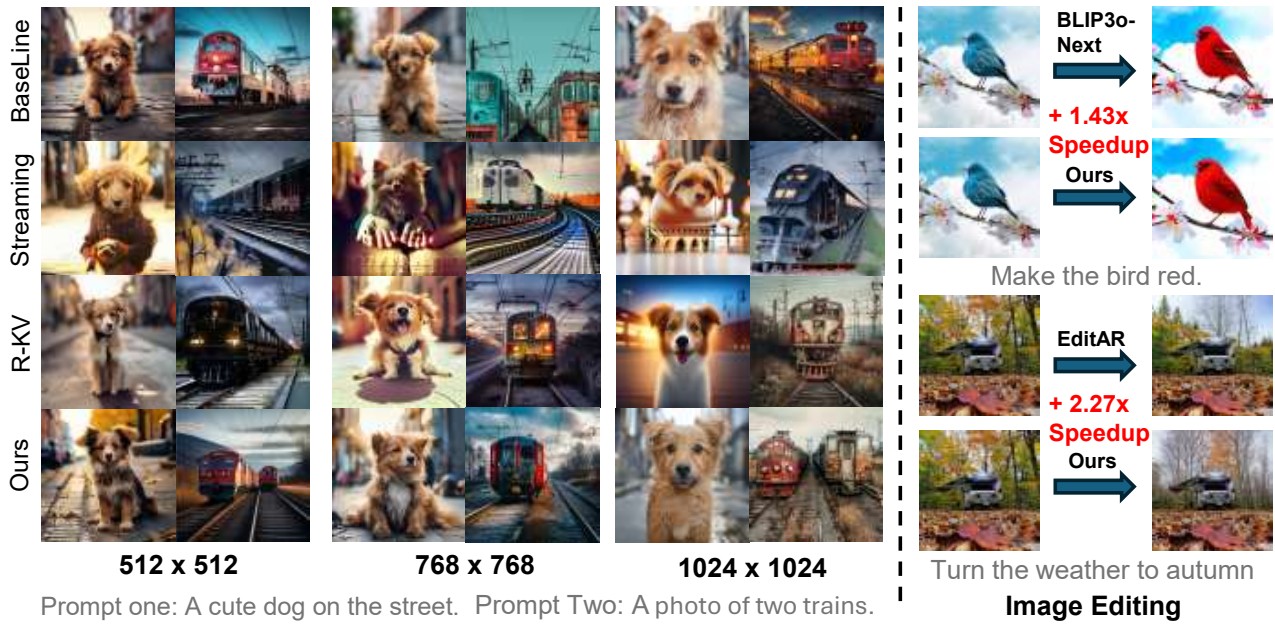

512 x 512 768 x 768 1024 x 1024 Image Editing

Prompt one: A cute dog on the street. Prompt Two: A photo of two trains.

*Figure 6.* **Left: Qualitative comparison of SparseAR with the other mainstream KV compression methods based on the Lumina-mGPT (Liu et al., 2024b) model series for text-to-image generation.** By comparing the generation results at multiple resolution scales, it can be seen that the proposed method has advantages in maintaining semantic consistency and structural integrity. **Right: Comparison with fully autoregressive EditAR (Mu et al., 2025) and hybrid AR-diffusion BLIP3o-Next (Chen et al., 2025b)** on image editing, where SparseAR achieves x% inference speedup while maintaining comparable visual fidelity.

ated later increasingly more defined. Simply put, the earlier the sparse attention mode is enabled, the more significant the impact on the generated results may be.

As shown in the Figure 6, when the streaming kv compression method is applied to Lumina-mGPT generation at various resolutions, it leads to a significant decrease in image quality and weakened text conformity. For example, the human-shaped puppy in the first column and the distorted train generated at 768 resolution. Multiple results for SparseAR are close to BaseLine, and even more accurate results (number of trains) are generated. The quantitative results are shown in Table 1, on models like LlamaGen-XL and Lumina-mGPT-512 when the startup are set to 1/6 and 1/8, the proposed method achieves an average speedup of nearly 10s and almost a 2x throughput speedup per sample, with virtually no performance degradation. When applied to high-resolution image generation models such as Lumina-mGPT-768 and Lumina-mGPT-1024, SparseAR still maintains similar quality and a considerable speedup. It is worth noting that as $tau$ decreases, for example, $\tau = 1/8$ at 1024 resolution, the number of steps required to enable SparseAR differs by $\tau = 1/4$ (256 steps), yet the FID and CLIP scores remain similar. This demonstrates that our method achieves a **more significant trade-off between efficiency and quality.**

**Image Editing.** Recent works (Chen et al., 2025b;a; Mu

et al., 2025; Xin et al., 2025) has already demonstrated the potential of AR models in controllable image generation tasks. Our method can be further applied to mainstream models for these image-to-image tasks. For image editing tasks, we evaluated the proposed method against the baseline on ImgEdit-Bench (Ye et al., 2025b).

As shown in the Table 2, it can be seen that under the 1/4 and 1/6 settings of EditAR, SparseAR shows almost no loss in text adherence for add, delete, and count prompts, and all metrics remain virtually unchanged, even showing improvement in some aspects. Furthermore, unlike pure AR models such as Lumina-mGPT and EditAR, Blip3o-Next is a unified understanding and generation model that combines AR and Diffusion, demonstrating more powerful performance in image generation tasks. The results in the table also show that our method maintains quality and achieves significant efficiency improvements under these settings. This demonstrates the versatility of our method. Figure 6 also shows a comparison of the raw image results between SparseAR and Baseline. For the same image editing samples, our method achieves almost the same results while achieving a significant speedup.

### 4.3. Comparison of Different Sparsity Mode

Previous researches (Xiao et al., 2024b; Zhang et al., 2023; Li et al., 2024b; Qin et al., 2025a) has focused on evict-

*Table 1.* **Quantitative results of text-to-image generation on LlamaGen-XL, Lumina-mGPT variants with our method under different hyperparameter $\tau$.** The hyperparameter $\tau$ serves as the activation threshold, defining the number of initial decoding steps performed with full attention to preserve global structural coherence before transitioning to our sparse attention mechanism. We also compare our method SparseAR with LineAR (Qin et al., 2025a).

| Model | Config | $\tau$ | GenEval↑ | FID-30K↓ | CLIP Score↑ | Latency(s)↓ | Throughput(tokens/s)↑ |
|---|---|---|---|---|---|---|---|
| LlamaGen-XL (N=576) | BaseLine | - | **0.33** | 5.67 | 32.75 | 61.54 | 11.24 |
| | SparseAR | 1/6 | **0.33** | **5.55** | **32.66** | 55.38 | 12.43 |
| | SparseAR | 1/8 | 0.30 | 5.89 | 32.18 | **54.27** | **12.74** |
| Lumina-mGPT-512 (N=1024) | BaseLine | - | **0.51** | **6.49** | 30.21 | 126.35 | 9.37 |
| | SparseAR | 1/6 | 0.50 | 6.54 | **31.45** | 119.24 | 10.41 |
| | SparseAR | 1/8 | 0.49 | 7.22 | 30.06 | **116.71** | **10.62** |
| Lumina-mGPT-768 (N=2304) | BaseLine | - | 0.52 | 4.92 | 33.88 | 264.27 | 10.46 |
| | LineAR | - | 0.52 | - | - | - | - |
| | SparseAR | 1/4 | **0.55** | **4.59** | **34.19** | 245.34 | 11.58 |
| | SparseAR | 1/6 | 0.52 | 4.91 | 33.82 | 242.67 | 11.64 |
| | SparseAR | 1/8 | 0.48 | 10.88 | 29.44 | **241.21** | **11.90** |
| Lumina-mGPT-1024 (N = 4096) | BaseLine | - | 0.56 | **3.22** | 31.77 | 514.54 | 9.55 |
| | LineAR | - | 0.56 | - | - | - | - |
| | SparseAR | 1/4 | **0.58** | 3.41 | **32.34** | 501.33 | 11.19 |
| | SparseAR | 1/6 | 0.56 | 3.37 | 31.11 | 494.45 | 12.27 |
| | SparseAR | 1/8 | 0.55 | 4.08 | 30.28 | **491.32** | **12.48** |

*Table 2.* **Quantitative results for image editing on ImgEdit (Ye et al., 2025b)**, with all metrics evaluated using GPT-4.1. The "Overall" score is computed as the average across all task categories. The hyperparameter $\tau$ serves as the activation threshold, defining the number of initial decoding steps performed with full attention to preserve global structural coherence before starting our sparse attention mechanism.

| Model | Config | $\tau$ | Add↑ | Extract↑ | Replace↑ | Remove↑ | Overall ↑ | Lat. (s)↓ | Through. (bit/s)↑ |
|---|---|---|---|---|---|---|---|---|---|
| BLIP3o-Next (Chen et al., 2025b) | Baseline | - | 4.00 | 2.39 | **4.05** | 2.61 | **3.62** | 122.78 | 5.79 |
| | SparseAR | 1/4 | **4.11** | **2.42** | **4.05** | 2.61 | 3.60 | 111.74 | 5.98 |
| | SparseAR | 1/6 | 4.02 | 2.41 | 4.03 | **2.62** | 3.61 | **108.33** | **6.13** |
| EditAR (Mu et al., 2025) | Baseline | - | **2.51** | 2.07 | 2.82 | **2.03** | **2.21** | 82.37 | 13.26 |
| | SparseAR | 1/4 | 2.48 | **2.08** | **2.83** | **2.03** | 2.20 | 77.61 | 14.08 |
| | SparseAR | 1/6 | 2.49 | 2.05 | 2.80 | 2.00 | 2.15 | **76.33** | **14.31** |
| Lumina-mGPT (Liu et al., 2024b) | Baseline | - | 1.95 | **1.44** | **1.45** | 1.33 | **1.51** | 142.59 | 8.24 |
| | SparseAR | 1/4 | **1.99** | 1.43 | 1.28 | **1.39** | 1.49 | 133.71 | 8.31 |
| | SparseAR | 1/6 | 1.93 | 1.39 | 1.31 | 1.28 | 1.47 | **131.08** | **8.77** |

ing a portion of the KV cache during compression, thereby saving memory and accelerating inference. These methods are primarily applied to large language models. Due to the semantics of textual information are relatively uniform, these studies explore patterns in text tokens using a uniform approach. However, for visual generation, the semantics of visual features are highly uneven and contain significant redundancy (Ma et al., 2025a;b; Teng et al., 2025; Tan et al., 2025). As mentioned earlier, KV compression methods applicable to LLM significantly affect the quality of generated images for text-to-image generation. To investigate this phenomenon in image editing tasks, we applied these methods to the Blip3o-Next model for image editing task. As shown in the Table 3, Streaming and R-KV methods show significant decreases in metrics such as FIP and Imgedit, indicating that compression in these modes still has a significant impact on image generation. Moreover, our method compresses both the conditional image and the generation

process simultaneously, maintaining stable generation while significantly reducing FLOPS and latency.

### 4.4. Effect of Compression Ratio on Different Regions

The compression ratio of SparseAR can be controlled by enabling the step 'tau' and the compression budge (the number of KV caches compressed). To further investigate the performance of our method at different compression ratios (its tolerance for compression), we performed ablation experiments. Specifically, we tested the performance of various compression budgets under three different parameter values $tau$, which directly affect the KV compression strength, as described in section 4.2. As shown in the Figure 7, when the compression budget is set to 20%, LlamaGen performs slightly worse at $\tau = 1/6$ and $\tau = 1/4$, but performs better at $\tau = 1/8$. A similar situation occurs for Lumina-mGPT-768 when the compression budget is 40%.

*Table 3.* Quantitative comparison of KV cache compression methods for image editing tasks based on BLIP3o-Next. **B** represents the compression budget setting of the applied KV compression method. Imgedit represents the result of ImageEdit Benchmark (Ye et al., 2025b). For Latency, we calculate it using the time spent generating each image. We calculated FID scores using 5000 images sampled from the OminiEdit dataset (Wei et al., 2025). **Bold** indicates the best results and underline indicates the second-best results

| Method | B | $\tau$ | FID-5k↓ | Imgedit ↑ | Latency | FLOPs |
|---|---|---|---|---|---|---|
| **Base** | Full | - | **13.12** | **3.62** | 122.78s | 28.53 |
| **Streaming** | 1/6 | - | 20.61 | 0.27 | 101.56 | 22.34 |
| | 1/8 | - | 23.17 | 1.98 | **99.32s** | **20.66** |
| **H2O** | 1/6 | - | 19.55 | 2.20 | 107.23s | 25.67 |
| | 1/8 | - | 25.46 | 2.02 | 106.25s | 24.55 |
| **R-KV** | 1/6 | - | 17.63 | 2.66 | 113.52s | 25.32 |
| | 1/8 | - | 22.39 | 2.95 | 111.90s | 24.06 |
| SparseAR | - | 1/4 | 13.80 | **3.62** | 111.74s | 24.11 |
| | - | 1/6 | 14.06 | 3.54 | 105.53s | 23.28 |

This is because the autoregressive model acquires more explicit context and background information as the generation process progresses, becoming less sensitive to compression. Conversely, significant compression has a more pronounced impact when insufficient image information is generated.

*Table 4.* Distribution of generation steps across entropy regions.

| Model | Low entropy | Mid entropy | High entropy |
|---|---|---|---|
| Lumina-mGPT | 30.0% / 3.00M | 25.0% / 3.07M | 45.0% / 4.16M |
| Blip3o-Next | 32.0% / 3.67M | 22.0% / 4.16M | 52.0% / 6.51M |

## 5. Discussion

**Performance in Extreme Scenarios.** In most cases, AR models exhibit low entropy in simple background areas and high entropy in complex, detailed areas. However, we observed extreme scenarios in the generated images, such as over 80% of the tokens exhibiting high entropy, where SparseAR may have limitations. To more comprehensively evaluate the performance of our method in extreme scenarios and whether this reduces its general applicability, we conducted a statistical analysis experiment. We generated approximately 5000 images from datasets such as Laion-coco and GenEval using Lumina-mGPT-512 and calculated the proportions of high, medium, and low entropy images. The results are shown in the Table 4 (**M** indicates Million.). High-entropy regions constitute the largest proportion but are not absolutely dominant and a large number of samples still contain significantly low-entropy regions. This indicates that the SparseAR can work in most cases. Furthermore, this demonstrates that our method provides acceleration in the vast majority of cases, especially image editing tasks.

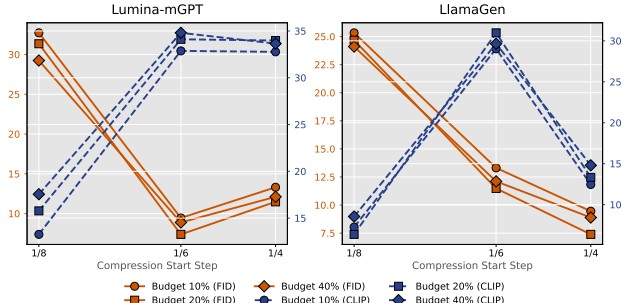

*Figure 7.* **Comparison of performance with SparseAR on Lumina-mGPT and LlamaGen under different compression budgets.** The x-axis represents the number of steps where SparseAR is enabled, and Budget Ratio represents the degree of sparsity when sparse attention is enabled. For example, a Budget Ratio of 10% means that 10 of unimportant tokens are removed in high-entropy regions, while only 10% of tokens are retained in low-entropy regions.

**Extension on Next-scale or Multi-scale AR Models.** SparseAR can be transferred to other generative architectures with only minor modifications. To begin, we demonstrate that the VAR (Li et al., 2025) and Meissonic (Bai et al., 2024) can also exhibit uncertainty patterns similar to raster-order AR. The visualization results are shown in Appendix E, which prove our point.

We hypothesized a simple implementation scheme. For VAR, it generates images scale-by-scale from low resolution. All tokens at the current scale share the cached KV from previous scales, so entropy can be estimated by mapping the entropy from the previous scale to the current scale. For Meissonic-like model, this approach uses bidirectional attention and generates tokens at completely random locations, prioritizing low-entropy locations. Therefore, we envision directly using sparse attention after several generation steps, as the model still expects to receive low-entropy tokens, thus allowing for attention computation.

## 6. Conclusion

In this paper, we purpose SparseAR, a novel training-free entropy-aware method tailored for visual autoregressive model. SparseAR is motivated by the fundamental observation that visual tokens exhibit strong spatial redundancy and highly non-uniform information density. Building on this insight, SparseAR leverages predictive entropy as a lightweight yet effective signal to dynamically distinguish information-rich and redundant regions during decoding, and adaptively allocates attention and KV cache budgets accordingly. SparseAR is plug-and-play, and applicable to both text-to-image generation and image editing across mainstream AR models. Extensive experiments demonstrate that it substantially improves inference efficiency while preserving, and often improving, generation quality.

## Acknowledgements

This project is supported by National Natural Science Foundation of China (grant number 62502544).

## Impact Statement

Our research improves the computational efficiency of autoregressive visual generation, fostering the development of more responsive and accessible creative tools. While the enhanced speed and quality of SparseAR have positive implications for digital art and design, we acknowledge that accelerated generative capabilities also heighten the importance of responsible AI deployment. We encourage the integration of our method with robust synthetic media detection and safety frameworks to ensure that the increased efficiency serves to benefit society while mitigating potential misuses such as the creation of misleading content.

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

# A. Implementation Details

## A.1. Baseline Compression Methods

Regarding the mainstream compression methods (Xiao et al., 2024b; Zhang et al., 2023) reported in the paper, such as Streaming, H2O, and R-KV, we directly apply the basic method flow described in the original paper to the visual generation model.

**Streaming.** Following the original paper, we first set the compression budget size. For example, 1/8 means that 1/8 of the total number of generated tokens is allocated to the initial $KV_{prefix}$ and the $KV_{local}$ closest to the generation position. Simultaneously, after reaching the required number, the $KV_{distant}$ parts excluding these two are discarded.

**H2O.** We collect and sum the attention weights received by each token during the generation process. By sorting these tokens according to their weights, when the preset compression budget is reached, we discard the last few tokens in the sorted list to ensure that the KV length remains within the compression budget.

**R-KV.** First, we use multi-head attention to comprehensively evaluate the contribution of each token. Second, we calculate the cosine similarity of the key vectors to identify "repeating" content. Finally, the KV cache is allocated in real-time according to the principle of high importance and low redundancy. A compression budget is used to control the degree of retention and discarding.

## A.2. Proposed Entropy-Aware KV Compression

For the proposed method, we follow the flow of Algorithm 1.

### A.2.1. NEIGHBOR COLLECTION

To decide what regions the current generation step belongs to, we collect causal temporal and spatial neighbors that are already generated:

$$T_i(w) = \{j : \max(1, i - w) \leq j < i\} \quad (2)$$
$$S_i(r) = \{j : |r(j) - r(i)| \leq r, \ |c(j) - c(i)| \leq r\} \quad (3)$$

where $w$ is the temporal window size and $r$ the spatial radius. $r(\cdot), c(\cdot)$ map linear position index to raster coordinates.

### A.2.2. LOCAL ENTROPY SCORE

We define the local entropy sample set as:

$$\mathcal{E}_i = \{e_j : j \in T_i(w) \cup S_i(r)\}$$

When $\mathcal{E}_i \neq \varnothing$, we compute a robust local score:

$$s_i = \text{median}(\mathcal{E}_i) \quad (4)$$

### A.2.3. REGION CLASSIFICATION

We classify the state of the current step $i$ according to the following rules:

$$C(i) = \begin{cases} \text{high entropy,} & s_i \geq threshold_{\text{high}}, \\ \text{normal,} & threshold_{\text{low}} < s_i < threshold_{\text{high}}, \\ \text{low entropy,} & s_i \leq threshold_{\text{low}}. \end{cases} \quad (5)$$

We determined the high and low entropy thresholds for each model by observing and statistically analyzing quantiles on 3K samples.

## A.3. Dynamic KV Cache Allocation

Based on the estimated entropy region of the current step $S_i$, as shown in Figure 13, we dynamically adjust the KV cache budget for attention computation.

**High-entropy regions.** Complex textures and structural details require extensive contextual information. We retain the full KV cache:

$$\mathbf{KV}_{\text{high}} = \mathbf{P}_{1:i} = \{\mathbf{K}_{1:i}, \mathbf{V}_{1:i}\} \quad (6)$$

**Low-entropy regions.** Uniform backgrounds and simple patterns allow a minimal attention approach, retaining only the initial prefix and recent local KV cache:

$$\mathbf{KV}_{\text{low}} = \mathbf{KV}_{\text{prefix}} \cup \mathbf{KV}_{\text{local}} \quad (7)$$

**Normal regions.** Inspired by Quest (Tang et al., 2024; Wu et al., 2025b), we use a query-based approach to select the top-K most important historical tokens for attention computation. We control the budget for $KV_{distant}$ by setting the size of $K$:

$$\mathbf{KV} \leftarrow \mathbf{KV}_{\text{prefix}}(P) \cup \mathbf{KV}_{\text{local}}(L) \cup \mathbf{KV}_{\text{distant}}^{\text{topk}}(S) \quad (8)$$

## A.4. Specific values of hyperparameters $\tau$

$\tau$ is used to indicate the number of steps to enable sparse attention. However, in the paper, for ease of expression, $\tau$ is expressed as a proportion of the total steps, not a fixed number. For example, with 1024 total steps for generating one image, $\tau = 1/4$ corresponds to step 256, meaning the module begins at the 25% mark of the sparse attention operation. We clarify the specific details regarding the parameter $\tau$ in Table 5.

## A.5. Threshold Selection for Entropy Regions

To determine appropriate entropy thresholds for region classification, we aggregated entropy values from a total of

*Table 5.* Step Index and hyperparameters $\tau$

| Model | $\tau$ | Step Index | Total Steps |
|---|---|---|---|
| LlamaGen-XL | 1/8 | 72 | 576 |
| LlamaGen-XL | 1/6 | 96 | 576 |
| Lumina-mGPT-512 | 1/8 | 128 | 1024 |
| Lumina-mGPT-512 | 1/6 | 170 | 1024 |
| Lumina-mGPT-768 | 1/8 | 288 | 2304 |
| Lumina-mGPT-768 | 1/6 | 384 | 2304 |
| Lumina-mGPT-1024 | 1/8 | 512 | 4096 |
| Lumina-mGPT-1024 | 1/6 | 682 | 4096 |

30,000 samples and categorized them into specific ranges based on empirical analysis. Our observations reveal that regions exhibiting high sensitivity in the entropy heatmap generally correspond to entropy values greater than 6.5, which roughly aligns with the 60th percentile of the entropy distribution. Conversely, low-entropy regions generally correspond to the 30th percentile. This pattern suggests a natural separation point for distinguishing between high- and low-entropy content regions. The specific results are presented in the table below, showcasing the entropy value distributions for different models.

*Table 6.* Entropy Threshold Ranges for Different Models

| Model | Low Entropy | Mid Entropy | High Entropy |
|---|---|---|---|
| EditAR | $(0, 3]$ | $(3, 6]$ | $(6, 8]$ |
| BLIP3o-Next | $(0, 2]$ | $(2, 5.5]$ | $(5.5, 7]$ |
| LlamaGen-XL | $(0, 2]$ | $(2, 7]$ | $(7, 8]$ |
| Lumina-mGPT-512 | $(0, 3]$ | $(3, 6.5]$ | $(6.5, 8]$ |

## B. Effect of Fixed-Pattern Compression Strategy on Generation Quality

As mentioned in the main text, based on our findings, we further designed a simple fixed-pattern compression strategy. Specifically, we observed that during autoregressive visual generation, attention often focuses on conditional and local contextual information. Following a simple intuition, we instructed the model to focus only on the conditional part and its adjacent image tokens during generation, using a parameter to control the number of these parts. For example, setting the parameter to 1/8 means using 1/8 of the KV Cache length of the generated tokens to select local KV for attention computation. We present a comparison between the results of this simplified compression and the baseline results in the Figure 9. The results demonstrate our view that simple compression leads to information loss.

## C. Discussion

### C.1. Locally Adjacent Similarity

**Entropy similarity of locally adjacent regions** As mentioned in 3 in the main text, we experimentally demonstrate that in most generation scenarios, locally adjacent locations exhibit entropy convergence and high similarity. To further illustrate this point, as shown in Figure 8 we show the correlation of specific entropy values at adjacent locations for more samples under the Lumina-mGPT-512 model.

**Facilitating text-to-image generation in long text scenarios** This paper proposes a KV compression method and achieves good performance on tasks such as text-to-image generation and image editing. Furthermore, we observe that SparseAR exhibits better text following ability than the baseline in some samples. Especially when the text instructions are long, we believe this is because existing mainstream models still perform poorly in long text scenarios (Jiao et al., 2025). Our method shortens the KV Cache, allowing the model to focus more on conditional information and thus more accurately follow the instructions.

**Extending on more conditional image generation tasks** In section 3.1, we found that autoregressive visual generation models, when performing image editing tasks, tend to focus their self-attention pattern on local and corresponding source image region conditional information. Further, following previous work (Xin et al., 2025; Mu et al., 2025; Tan et al., 2025), we initially tested the performance on a canny-to-image task. The generated results were promissory, as shown in Figure 10 Therefore, we believe that autoregressive visual generation for image-to-image tasks should all exhibit similar attention patterns to image editing, and we plan to further explore this in the future.

## D. Limitation

**Limited efficiency impact in bad scenarios.** Our method can identify potentially highly variable regions and redundant flat regions; however, as shown in Fig. 15, the proposed entropy estimation method does not perform well in certain situations. This same problem exists for model-based entropy estimation methods. For example, when encountering regions where the entire image is high-entropy or normal, our method applies query-based sparse attention to all regions or uses almost no constrained context. In such cases, the efficiency improvement offered by our method is limited.

## E. Additional Visual Comparison of Generation

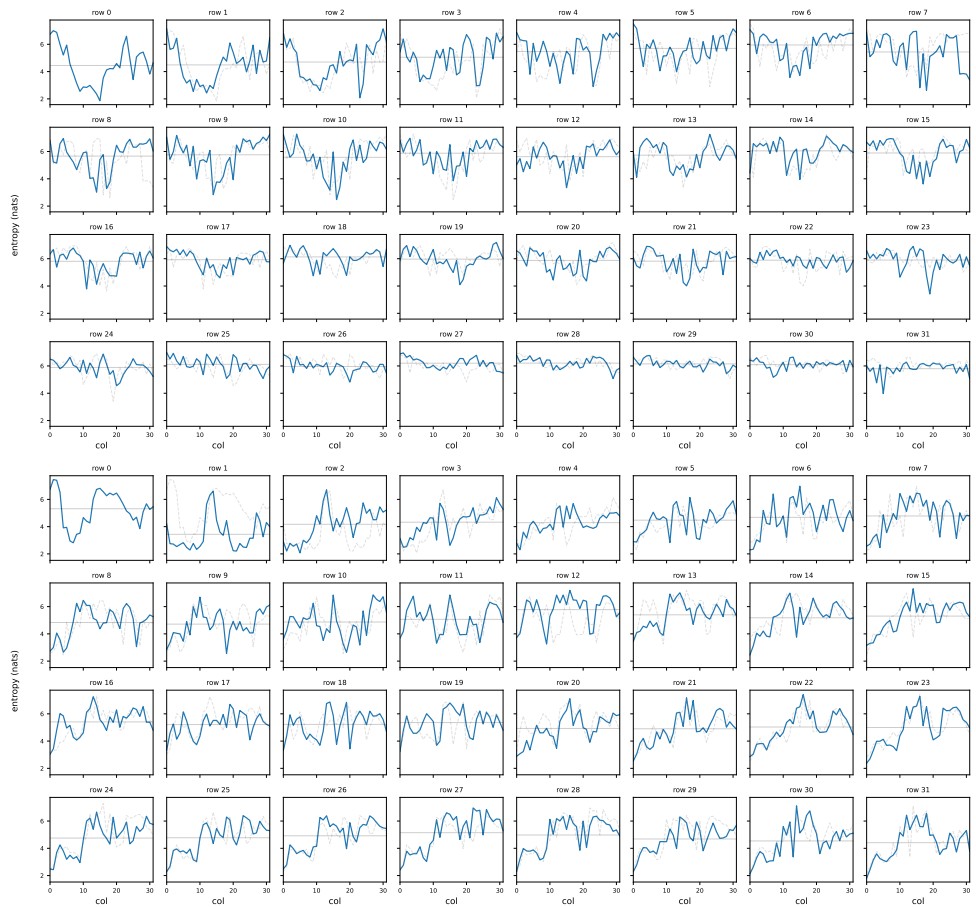

*Figure 8.* **Adjacent rows exhibit similar entropy curves from text-to-image generation of Lumina-mGPT.** This demonstrates the effectiveness of our proposed entropy estimation method. The light white dashed line represents the entropy curve of the previous row, and the blue curve is the entropy curve of the current row.

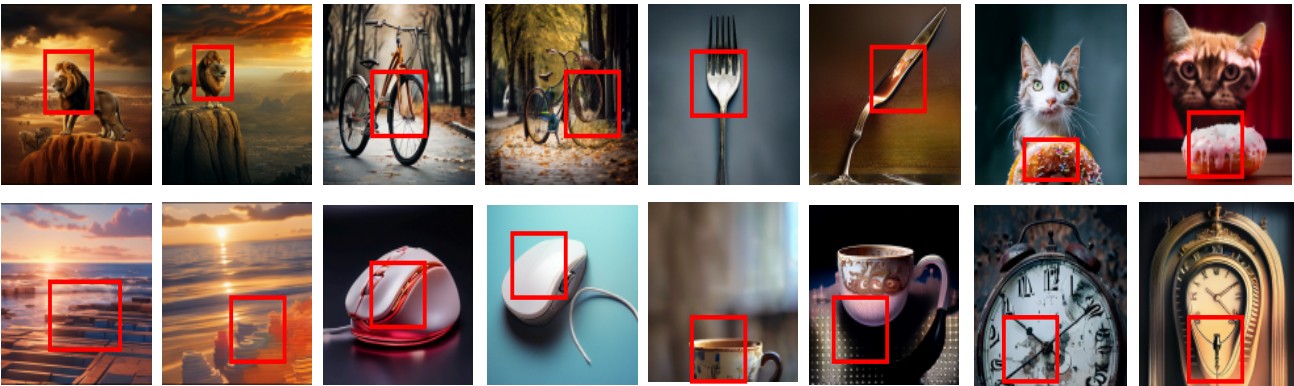

*Figure 9.* **Effect of position-based KV compression on autoregressive image generation.** The left side of each set of comparison images is the generated image of the baseline, and the right side is the generated image under the simple compression strategy.

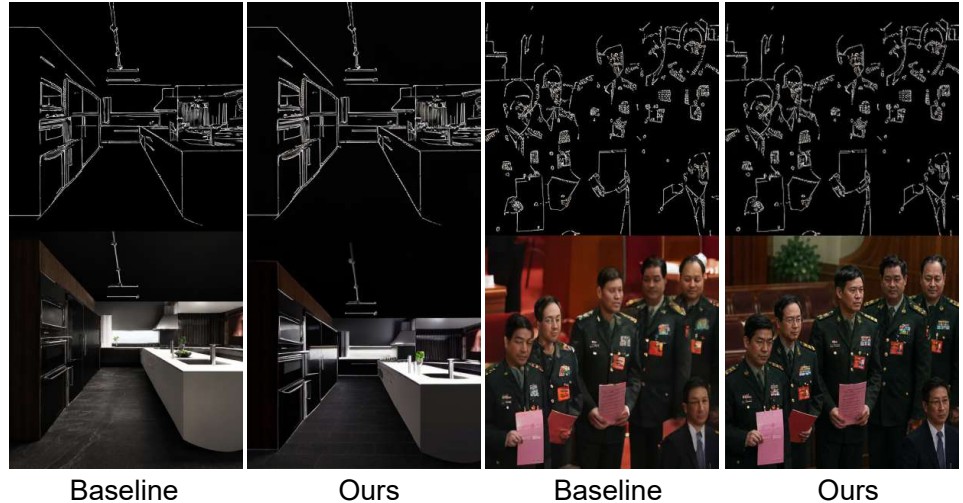

Baseline        Ours        Baseline        Ours

*Figure 10.* Comparison of Canny-to-Image Generation Results.

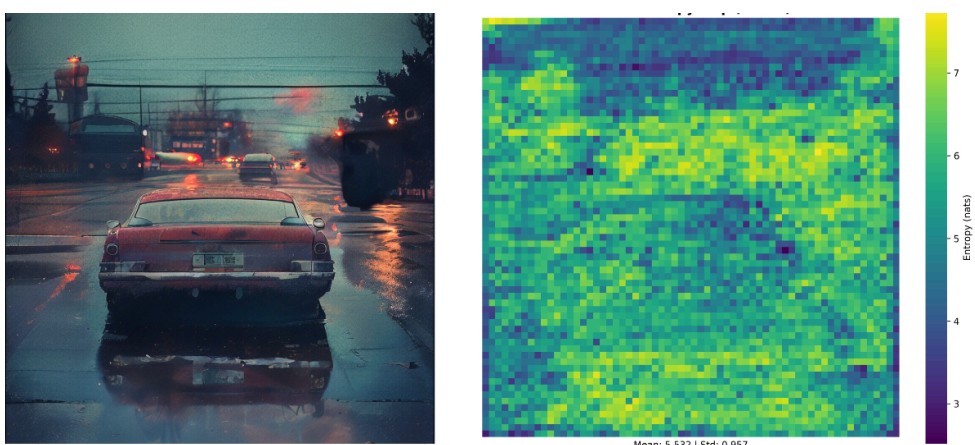

*Figure 11.* scenarios where entropy maps perform poorly for SparseAR.

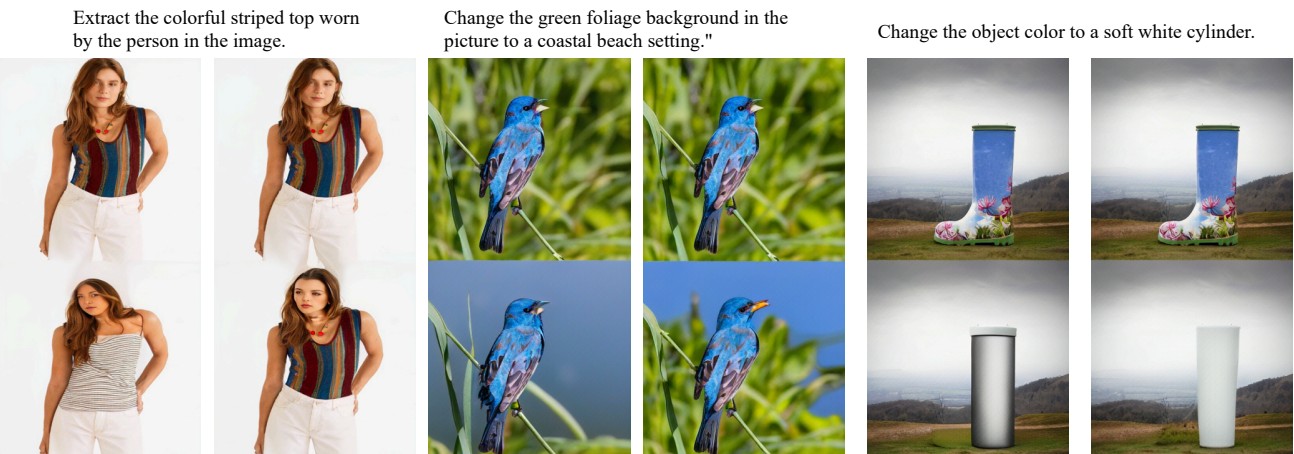

*Figure 12.* **Visualization of proposed SparseAR on Blip3o-Next for image editing**.

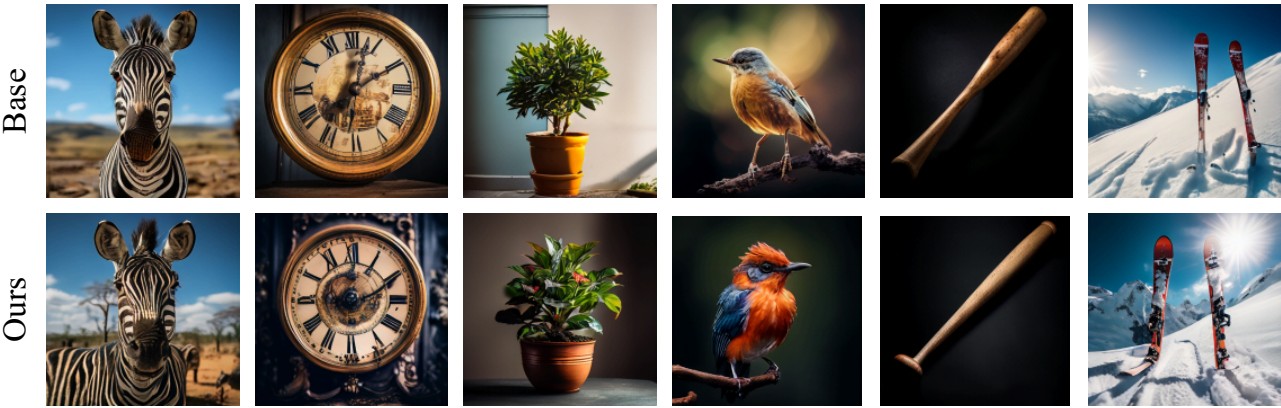

*Figure 13.* **Visualization of proposed SparseAR on Lumina-mGPT-512 for text-to-image generation**.

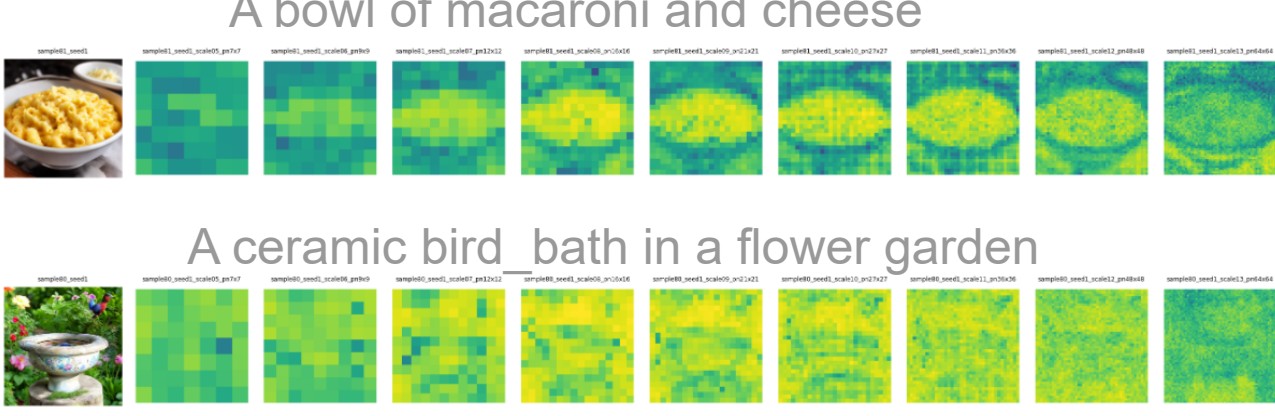

*Figure 14.* Entropy Map of VAR-like Model for text-to-image generation.

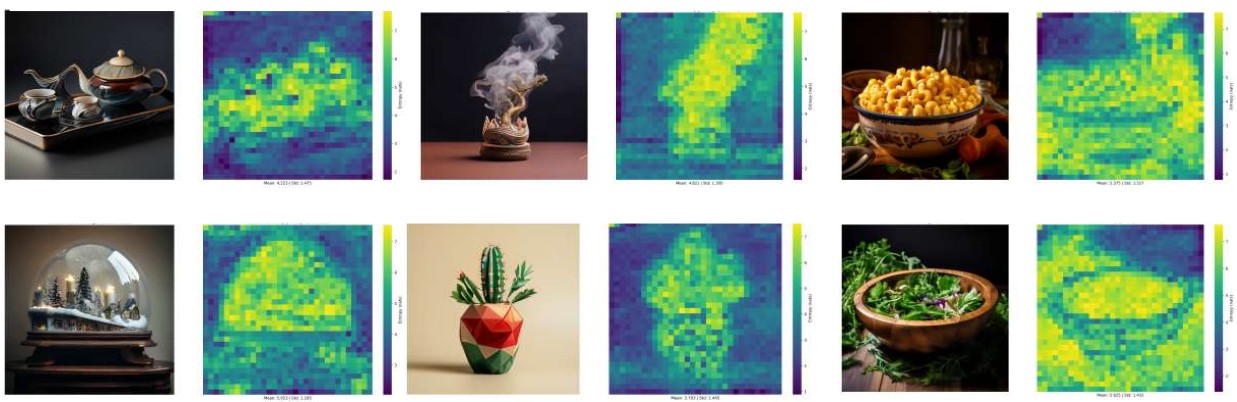

*Figure 15.* More Entropy Map Visualization of proposed SparseAR on Lumina-mGPT-512 for text-to-image generation.

