# OpenReview forum: "Entropy-Aware Dynamic KV Cache Sparsification for  Autoregressive Image Generation and Editing"
_ICML.cc/2026/Conference — ICML 2026 regular_

### Official Review · Reviewer_kURW · 2026-02-17

**Soundness:** 2
**Presentation:** 2
**Significance:** 1
**Originality:** 2
**Overall Recommendation:** 3
**Confidence:** 4

**Summary:**

### Summary
This paper proposes a method to reduce the KV cache memory requirements of Autoregressive (AR) image models based on next-token prediction. Specifically, the paper demonstrates that during AR image generation: (1) different patches (tokens) possess varying token entropy, (2) locally adjacent regions exhibit similar entropy levels, and (3) sparsifying the KV cache when generating low-entropy tokens does not lead to a substantial drop in performance. Based on these insights, the paper introduces a method to dynamically adjust KV cache sparsity according to token-entropy Experimental results show that the proposed approach significantly reduces the memory demands of AR models and improves latency without compromising generation quality.

**Compliance With Llm Reviewing Policy:**

Affirmed.

**Final Justification:**

My main concerns still remain. (WK3) LLMs also face large-scale decoding issues, and current autoregressive image models do not operate at such long token lengths. Although the authors assume some multi-turn editing scenario in their rebuttal, the main paper does not include experiments on such a setting.  (WK1) I am still not convinced that this sufficiently justifies the effectiveness of using entropy. Moreover, since the initial draft did not include such discussion about the paper’s main motivation, I find it difficult to revise my evaluation based on it.

However, considering the authors’ response, I will raise my score to 3.

**Key Questions For Authors:**

Please see the weaknesses above.

Also, I recommend revising the paper to include a more detailed analysis of why entropy serves as an effective metric for sparsification and including a detailed layer/head-wise analysis of the impact of KV cache sparsification. This will significantly strengthen the paper's contribution.

**Limitations:**

Yes

**Strengths And Weaknesses:**

### Strengths
- The paper is well-written and easy to follow.
- The proposed method is simple yet delivers strong performance, especially in throughput.

### Weaknesses
- **Novelty**: The method essentially proposes using a high sparsity ratio for background areas and a low sparsity ratio for object regions. It lacks new mechanisms or discoveries, which limits its academic novelty.
- **Ambiguity of the Entropy Metric**: Asserting entropy as the primary metric for sparsity throughout the paper is not sufficiently justified and relies solely on the empirical evidence presented in Fig. 3(b).
    1. As shown in Appendix Fig. 9, token entropy is not always strictly correlated with the background/object. Fig. 3(a) may be cherry-picked.
    2. While the paper states that background regions like the sky contain many similar tokens (Line 361), it claims that these regions have low entropy (Line 185), which seems contradictory.
- **Motivation**: The main motivation of this paper requires some reconsideration. Unlike Text LLMs, which usually require 100k to millions of tokens and thus face massive KV cache issues, Image AR models typically handle tokens in the range of thousands. The performance gain may be limited unless we assume a very large batch size.
- **Minor**: The reference to "Appendix 4.3" in Line 357 appears to be incorrect as it does not exist.

---

> ### Author Rebuttal · Authors · 2026-03-28
>
> >**"WK1: The method lacks new mechanisms or discoveries, which limits its academic novelty."**
>
> KV compression or sparsification has **been extensively studied** within the research of Large Language Models.
> However, due to the **fundamental differences** between visual and textual tokens, methods effective for LLMs **do not translate well** to visual autoregressive models.
> Specifically, if KV compression is applied to highly complex regions, the resulting shortened context can lead to erroneous generated outputs, thereby compromising the integrity of all subsequent generated results. To address this issue, we use the model's prediction entropy to indicate information density and dynamically implement KV sparsification.
>
> To more clearly illustrate our contributions, we present a comparison with other classic methods in the following table.
>
> | Method | Large Language Model | Class-to-Image | Text-to-Image | Image Editing |
> |:---:|:---:|:---:|:---:|:---:|
> | **Speculative Decoding** | FastSD[1] | — | SJD[2], Hawk[3] | — |
> | **Multi-Token Generation** | Medusa[4] | LPD[5], RandAR[6] | Token-shuffle[7] | — |
> | **KV Cache Compression** | StreamingLLM[8], Quest[9] | — | LineAR[10], SparseAR | SparseAR |
>
> ` As shown in the table,  the proposed SparseAR innovatively explores the KV Cache problem of the AR image generation model and proposes a compression method unique to the visual AR model, especially image editing. `
>
> >**"WK2.1: As shown in Appendix Fig. 9, token entropy is not always strictly correlated with the background/object. Fig. 3(a) may be cherry-picked."**
>
> **Figure 9 presents a set of examples illustrating extreme scenarios, rather than typical ones.** To substantiate this point, we conducted an experiment: we used Lumina-mGPT-512 and Blip3o-Next to quantify the entropy statistics for a sample of 5,000 images and plotted the results as a histogram. As the result illustrates, although tokens with high entropy constitute a larger proportion of the data, they outnumber low-entropy tokens by **only approximately 10%.** This statistic demonstrates that **Figure 3(a) was not "cherry-picked."**
>
> We present the specific quantitative results in the table below: The percentage and the specific number of tokens for each entropy category when generating 5000 samples.
> The results are visualized in the images accessible via the following link: **[Anonymous Github Link](https://anonymous.4open.science/r/tt_tt_374698/.)**
>
> | Models| Low entropy | Mid entropy | High entropy |
> |:---:|:---:|:---:|:---:|
> | Lumina-mGPT (Text-to-Image) |29.3% (2,999,719) | 30.0% (3,071,385) | 40.6% (4,156,608) |
> | Blip3o-Next (Image Eding) |32.8% (3,674,983) | 25.9% (4,163,277) | 48.2% (6,511,329) |
>
> ` The results in the figure show that high-entropy tokens are numerous but not overwhelmingly dominant. `
>
> >**"WK2.2: While the paper states that background regions like the sky contain many similar tokens (Line 361), it claims that these regions have low entropy (Line 185), which seems contradictory."**
>
> In this paper, we design our methods and experiments based on the idea that high entropy and low entropy can indicate **complex and simple regions, respectively.** Therefore, the fact that background regions **contain many similar tokens** and exhibit low entropy **aligns with our opinions.**
>
> >**"WK3: The main motivation requires some reconsideration. Unlike Text LLMs, which usually require 100k to millions of tokens and thus face massive KV cache issues, Image AR models typically handle tokens in the range of thousands."**
>
> **Visual tokens constitute a significant portion of the input in large multimodal models.** Specifically, autoregressive models such as Lumina-mGPT, Janus-Pro, and Emu3.5 are often models that **unify generation and understanding,** capable of simultaneously answering questions and generating images or videos. A characteristic of these models is that they often have significantly more image tokens than text tokens. For example, the Lumina-mGPT-1024  typically processes text tokens of up to 80 characters in length, yet generates as many as 4096 image tokens. **Therefore, we propose that token efficiency is necessary for these models.**
>
> >**"WK4: The reference to "Appendix 4.3" in Line 357 appears to be incorrect as it does not exist."**
>
> Line 357 is a typo and thank you for pointing it out.
>
> **References:**
>
> [1] [FastSD](https://arxiv.org/abs/2211.17192). ICML 2023
>
> [2] [SJD](https://arxiv.org/abs/2410.01699). ICLR 2025
>
> [3] [Hawk](https://arxiv.org/abs/2510.25739). NeurIPS 2025
>
> [4] [Medusa](https://arxiv.org/abs/2401.10774). ICML 2024
>
> [5] [LPD](https://arxiv.org/abs/2507.01957). ICLR 2026
>
> [6] [RandAR](https://arxiv.org/abs/2412.01827). CVPR 2025
>
> [7] [Token-shuffle](https://arxiv.org/abs/2504.17789).
>
> [8] [StreamingLLM](https://arxiv.org/abs/2309.17453). ICLR 2024
>
> [9] [Quest](https://arxiv.org/abs/2406.10774). ICLR 2024
>
> [10] [LineAR](https://arxiv.org/abs/2512.04857).

---

> > ### Author Rebuttal · Reviewer_kURW · 2026-04-03
> >
> > Thanks for the rebuttal. However, the authors’ response does not resolve my concerns. (WK1) Why entropy is a useful metric for selecting unimportant tokens in visual AR is still not justified by the new experiment. (WK3) 4K tokens is not a large context length compared to text LLM, which handles ~ 100K tokens. AR image gen models usually generate only 1 image, so they typically do not suffer from severe KV-cache memory issues.

---

> > > ### Author Response · Authors · 2026-04-05
> > >
> > > Thank you for your continued feedback. As our earlier response was limited in scope, we would like to take this chance to provide a more comprehensive clarification.
> > >
> > > > **"Q1: Why entropy serves as a useful metric for selecting unimportant tokens in visual AR?"**
> > >
> > > `We observe that logit entropy in visual AR models reflects image information density (simple/complex content). `
> > >
> > > Building upon this observation, we conducted the experiments illustrated in Fig. 3(b) to demonstrate how varying entropy levels and compression ratios impact the quality of generated images. **The results demonstrate that performing compression within low-entropy regions has a negligible impact on generation quality, which constitutes our core hypothesis.**
> > >
> > > To futhure show entropy can effectively indicate redundant regions to a certain extent, we conducted two types of experiments: (a) Comparision with Powerful MLLM; (b) Comparison of Cosine Similarities among Image Tokens.
> > >
> > > (a) We selected 2,000 samples from Geneval [1] and used Lumina-mGPT-512/768 to generate outputs, recording logit entropy for each image token. Each image was then fed into Qwen2.5-VL [2]. "Region Match" measures the correspondence between low-entropy regions and high-redundancy regions/low attention regions.
> > >
> > > **First,** we input the prompt "Describe the object." and record the final-layer attention weights of all image tokens. For the 32×32 token image from Lumina-mGPT-512, we apply a 3×3 window-based partitioning and compute the ratio of low-entropy regions to low-attention tokens by comparing window-averaged entropy with token attention weights.
> > >
> > > | Low Entropy |  Region Match |
> > > | --- |:---:|
> > > | Lumina-mGPT-512 | 0.71 |
> > > | Lumina-mGPT-768 | 0.69  |
> > >
> > > Then, we directly compute the cosine similarity for all tokens generated by Qwen2.5-VL for each image. We then compare the proportion of tokens exhibiting low entropy and high cosine similarity (indicating high redundancy). If this proportion is high, it validates the **efficacy of using entropy** as an indicator.
> > >
> > > | Low Entropy |  Region Match |
> > > | --- |:---:|
> > > | Lumina-mGPT-512 | 0.63 |
> > > | Lumina-mGPT-768 | 0.71  |
> > >
> > > (b) In decoder-only autoregressive (AR) models, Our approach entails—once the generation of a complete image is finished—iterating through the sequence of image tokens using a sliding square window to calculate cosine similarities. If the similarity is high, it suggests that the region consists of simple, homogeneous pixels. Then, we further compare the average entropy values ​​within these windows to verify that regions exhibiting redundancy indeed correspond to low-entropy areas.
> > >
> > > | Model | Region Match |
> > > |---|:--:|
> > > | Lumina-mGPT-512 | 0.89 |
> > > | Blip3o-Next  | 0.85  |
> > >
> > > > **"Q2: 4K tokens is not a large context length compared to text LLM, which handles ~ 100K tokens. AR image gen models usually generate only 1 image, so they typically do not suffer from severe KV-cache memory issues."**
> > >
> > > We believe this comment conflates two fundamentally different inference workloads and we address this from two perspectives.
> > >
> > > ### ① Different bottleneck regime (prefill-dominated vs. decode-dominated).
> > >
> > > The 100K-token LLM comparison conflates prefill and decode workloads. In long-context LLM, the KV cache is built once via parallel prefill and accessed only briefly during short generation. In AR image generation, all 4K tokens are **decoded autoregressively,** the KV cache is read in full at every single decoding step, making it a purely decode-dominated bottleneck with no prefill to amortize. **Therefore, visual AR generative models do indeed suffer from KV cache issues.**
> > >
> > > Furthermore, the time gap between LLM and AR MLLM is large. As mentioned in Uncomp [3], in a scenario with 3712 prompts + 384 generate, Qwen2.5-7B's prefill phase took **77.34 seconds,** while decoding only took **51.79 seconds,** with prefill accounting for approximately **60% of the total inference time.** For Lumina-mGPT-1024, generating an image only takes **a fraction of a second** for prefill, but it takes **nearly 400 seconds** to generate 4K tokens.
> > >
> > > ### ② Long-context visual generation scenarios (Multi-turn and Multi-condition Image Editing/Generation)
> > >
> > > With the rapid advancement of  multimodal large models, image generation and editing have become exceptionally powerful—GPT-4o, for instance, can perform multi-round generation and editing/refinement seamlessly. This drives critical demand for multi-round editing and multi-conditional generation. In Lumina-mGPT 2.0 [4], a single 768×768 image requires **~9,000 tokens,** while multiple condition images exceed **20,000 tokens**—all stored as KV cache in GPU memory **(about 20 GB memory).**
> > >
> > > **This imposes massive memory pressure on visual autoregressive models, validating our research motivation.**
> > >
> > > **References:**
> > >
> > > [1] https://arxiv.org/abs/2310.11513
> > >
> > > [2] https://arxiv.org/abs/2310.11513
> > >
> > > [3] https://arxiv.org/abs/2410.03090 EMNLP 2025
> > >
> > > [4] https://arxiv.org/abs/2507.17801

---

### Official Review · Reviewer_XHyz · 2026-03-07

**Soundness:** 3
**Presentation:** 3
**Significance:** 3
**Originality:** 3
**Overall Recommendation:** 4
**Confidence:** 4

**Summary:**

This paper proposes SparseAR, a training-free and plug-and-play method to reduce KV-cache overhead for autoregressive image generation and editing. The main idea is to use next-token predictive entropy as a proxy for local information density: low-entropy (redundant/flat) regions can tolerate aggressive sparsification, while high-entropy (information-rich) regions benefit from keeping a larger context. SparseAR applies full attention for an initial warmup period and then dynamically classifies steps into low/normal/high-entropy regimes, allocating KV budget accordingly (full KV for high entropy, prefix+local KV for low entropy, and a top-K selection from distant KV for normal entropy). Experiments across multiple AR models and benchmarks show improved inference efficiency while largely maintaining (and sometimes improving) generation/editing quality.

**Compliance With Llm Reviewing Policy:**

Affirmed.

**Final Justification:**

My final justification is remaining the original score.

**Key Questions For Authors:**

1. Figure 3(b) protocol clarification. In Fig. 3(b), what exactly do the x-axis percentages represent (KV retention vs pruning ratio), and how is compression applied with respect to the entropy bins (e.g., compress only tokens whose entropy falls in
[0,3] while leaving others uncompressed, or something else)? A clear description (and ideally pseudocode) would help; this affects how strongly I buy the key empirical claim that “low-entropy regions are safe to compress.”

2. τ as ratio vs step index. Table 1 reports τ as {1/4,1/6,1/8} while Algorithm 1 treats τ as an “activation step.” Is τ implemented as 𝜏=⌊𝛼𝑁⌋
 where 𝑁=𝐻×𝑊, or as an absolute token index? Clarifying this affects how one would port the method across resolutions/models.

3. How are 𝜃_𝑙𝑜𝑤,𝜃_ℎ𝑖𝑔ℎ chosen exactly? The appendix mentions quantile-based selection over 3K samples, but does not state the quantiles or resulting numeric thresholds per model/resolution. Please report the quantiles and the actual threshold values used in experiments; this affects reproducibility and whether thresholds generalize across datasets/prompts.

4. Relation to LM uncertainty/entropy-aware KV selection. Since uncertainty-/entropy-aware and query-aware KV selection/compression exists for LMs, can you clarify what you see as the primary non-trivial/vision-specific contribution beyond adaptation (e.g., spatial correlation exploitation, region-wise policy, conditional token handling in editing)? A sharper positioning would strengthen the novelty claim.

**Limitations:**

yes

**Strengths And Weaknesses:**

Soundness:

The method is technically plausible: it uses next-token predictive entropy to categorize decoding steps into low/normal/high-entropy regimes and allocates KV budget accordingly (full KV for high entropy, prefix+local for low entropy, and top-K selection from distant KV for normal entropy). The overall design is consistent with the empirical claim that low-entropy regions are more redundant and less sensitive to long-range context.

However, some methodological details are under-specified for full reproducibility: Algorithm 1 introduces KV_topk^distant(S) without clearly defining the importance scoring/selection rule at first use, and threshold selection for 𝜃_low, 𝜃_high is described only qualitatively (quantiles over 3K samples) without exact quantiles or resulting numeric values.

Presentation:

The core intuition is easy to follow, but several clarity issues make the paper harder to read than necessary. In particular, Figure 3(b) is confusing: it mixes entropy bins (e.g., [0–3], [3–5]) with percentage labels, and it is unclear from the figure alone whether the percentages denote KV retention vs pruning and what the exact experimental protocol is (compress only tokens in a bin vs evaluate conditioned on entropy, etc.). Since this figure supports the key motivation, it should be clarified with explicit axis labels/legend and a short protocol description.
Notation would benefit from tightening: τ is used as an “activation step” in Algorithm 1, while Table 1 reports τ as a ratio (1/4, 1/6, 1/8). Defining 𝜏=⌊𝛼𝑁⌋ would remove ambiguity and avoid confusion with entropy thresholds 𝜃_low, 𝜃_high.

Significance:

The paper targets a practical bottleneck in AR image generation/editing (KV cache and attention cost for long visual token sequences). A training-free, plug-and-play approach that improves latency/throughput while maintaining quality is potentially valuable for deployment and for scaling AR visual generators to higher resolutions.

The reported speedups and multi-model coverage suggest the method could be useful in practice, although the strongest impact depends on how robust the entropy-based regime classification and top-K selection are across prompts, datasets, and resolutions.

Originality:

The high-level principle—using uncertainty/entropy signals to guide dynamic KV selection/compression—has precedent in language-model inference literature, and the paper would benefit from explicitly positioning against that line of work.

That said, applying the idea to visual AR decoding is not a pure copy-paste: the paper’s framing around spatially non-uniform redundancy, region-wise policies, and the observed local entropy correlation are vision-specific and help justify the adaptation. Overall, I view the novelty as incremental but useful, which aligns with a weak accept if the empirical gains are solid and the presentation issues are addressed.

---

> ### Author Rebuttal · Authors · 2026-03-29
>
> We sincerely appreciate the valuable time and extra effort you have dedicated to the review process. We value your remark that **"The paper targets a practical bottleneck in AR image generation/editing."** We appreciate your acknowledgment that this method are **"A training-free, plug-and-play approach that improves latency/throughput while maintaining quality."**
>
> >**"WK1: Figure 3(b) protocol clarification. In Fig. 3(b), what exactly do the x-axis percentages represent (KV retention vs pruning ratio), and how is compression applied with respect to the entropy bins?"**
>
> Regarding Figure 3(b), we selected 3,000 input samples and conducted experiments across the various entropy ranges listed. To assess the impact of compression on different entropy regions, we adopted the methodology proposed in streamLLM [1], applying KV compression to tokens falling within **specific entropy ranges**. Specifically, during the model's generation process, we designated a target entropy range; whenever the generation steps **entered this range,** we retained only a predetermined number of `prefix_tokens` and `recent_tokens`. By controlling the quantity of these retained tokens, we achieved **varying compression rates**. The entropy values ​​for real-time generated positions were determined using an auxiliary AR model, which we selected for this purpose.
>
> >**"WK2: τ as ratio vs step index. Table 1 reports τ as {1/4,1/6,1/8} while Algorithm 1 treats τ as an “activation step.” Is τ implemented as 𝜏=⌊𝛼𝑁⌋ where 𝑁=𝐻×𝑊, or as an absolute token index?"**
>
> Regarding the hyperparameter $\tau$, we use it to indicate the number of inference steps.  (e.g., $\tau = 55$ means starting from step 55).
>
> >**“WK3: How are 𝜃_𝑙𝑜𝑤,𝜃_ℎ𝑖𝑔ℎ chosen exactly? The appendix mentions quantile-based selection over 3K samples, but does not state the quantiles or resulting numeric thresholds per model/resolution.”**
>
> We aggregated the entropy values ​​for a total of 30,000 samples and categorized them into specific ranges. We discovered that the regions exhibiting **high sensitivity** on the heatmap generally correspond to entropy values ​​**greater than 6.5**—roughly aligning with the 60th percentile—while low-entropy regions generally correspond to the 30th percentile. Since entropy characterizes content-dependent predictive uncertainty and **is inherently independent of resolution,** this methodology is generalizable to other models. The specific results are presented in the table below, showcasing the entropy value distributions for two distinct models.
>
> More visualization results are displayed at **[Anonymous Github Link](https://anonymous.4open.science/r/tt_tt_374698/.)**.
>
> | Models| Low entropy | Mid entropy | High entropy |
> |:---:|:---:|:---:|:---:|
> |Lumina-mGPT |29.3% (2,999,719) | 30.0% (3,071,385) | 40.6% (4,156,608) |
> |Blip3o-Next |32.8% (3,674,983) | 25.9% (4,163,277) | 48.2% (6,511,329) |
>
> | Models | Low entropy| Mid entropy | High entropy |
> |:---:|:---:|:---:|:---:|
> | EditAR | (0,3 ] | (3, 6 ] | (6, 8] |
> | Blip3o-Next | (0, 2] | (2, 5.5] | (5.5, 7] |
> | LlamaGen-XL | (0, 2] | (2, 7] | (7, 8] |
> | Lumina-mGPT-512 | (0, 3] | (3, 6.5] | (6.5, 8] |
> >**“WK4" The primary non-trivial/vision-specific contribution beyond adaptation of SparseAR ?”**
>
> We will outline the contributions of our proposed method by addressing several aspects. Our contributions are:
>
> - ***Novel KV Sparsification Method:*** (i) Regarding KV Cache compression in LLMs, the field **has been extensively studied**. However, when attempting to **directly transfer** these methods to visual autoregressive models, significant challenges arise—such as **the lack of dedicated inference acceleration frameworks** (e.g., vLLM) and a substantial decline in performance. (ii) This work pioneers a KV sparsification method specifically tailored for visual autoregressive models, effectively addressing the issue of excessive KV Cache consumption prevalent in tasks such as text-to-image generation and image editing. (iii) There is **currently no** "uncertainty-/entropy-aware" KV compression method for LMs.
> - ***Efficient Attention Mechanism for AR Generation:*** Although visual AR image generation models and LLMs exhibit similar attention patterns during the generation process, it remains **difficult to directly transfer** findings and methodologies from LLMs to visual AR models. This difficulty stems from the distinct training paradigms employed for text-to-image generation and text-guided editing, as well as the fundamental **differences between the two modalities.** We are among **the very first** to conduct an in-depth investigation into the attention patterns of AR generation models. Particularly in the context of **image editing,** we were **the first** to identify a specific correspondence between the generation position and the conditional source image, leveraging this insight, we designed an "entropy-aware" sparse attention mechanism that has yielded excellent results.

---

> > ### Author Rebuttal · Reviewer_XHyz · 2026-04-03
> >
> > 1. **τ mismatch:** Table 1 reports τ as {1/4, 1/6, 1/8}, but the rebuttal says τ is an absolute index (e.g., 55). How exactly was τ set in Table 1 for each resolution/model? Please report the *actual step indices* used (per model/resolution), and update the paper to avoid ambiguity.
> >
> > 2. **Fig. 3(b) percentages:** Do the x-axis percentages denote **KV retention ratio** (kept tokens / original tokens) or **pruning ratio** (removed tokens / original tokens)? Please add a one-line definition in the caption.
> >
> > 3. **Entropy via auxiliary AR model:** In Fig. 3(b), entropy is computed “using an auxiliary AR model.” Is this auxiliary model the same architecture/checkpoint as the generator, or a different model? How sensitive are the conclusions to this choice?
> >
> > 4. **Novelty claim about LMs:** The rebuttal states there is no uncertainty-/entropy-aware KV compression for LMs. Can the authors verify this claim and either cite related LM work or revise the statement and clarify the novelty as a vision-specific adaptation?

---

> > > ### Author Response · Authors · 2026-04-05
> > >
> > > Thank you for your continued feedback. As our earlier response was limited in scope, we would like to take this chance to provide a more comprehensive clarification.
> > >
> > > >**"Q1: How exactly was τ set in Table 1 for each resolution/model?"**
> > >
> > > A1: **τ is used to indicate the number of steps to enable sparse attention.** As mentioned before, τ = 55 means starting from step 55. However, in the paper, for ease of expression, τ is expressed as a proportion of the total steps, not a fixed number. For example, with 1024 total steps for generating one image, τ = 1/4 corresponds to step 256, meaning the module begins at the 25% mark of the sparse attention operation. We clarify the specific details regarding **the parameter τ in Table 1.**
> > >
> > > | Model | τ | step index | total steps |
> > > |:---|:--|:---:|:----:|
> > > | LlamaGen-XL | 1/8 | 72 | 576 |
> > > | LlamaGen-XL | 1/6 | 96 | 576 |
> > > | Lumina-mGPT-512 | 1/8 | 128 | 1024 |
> > > | Lumina-mGPT-512 | 1/6 | 170 | 1024 |
> > > | Lumina-mGPT-768 | 1/8 | 288 | 2304 |
> > > | Lumina-mGPT-768 | 1/6 | 384 | 2304 |
> > > | Lumina-mGPT-1024 | 1/8 | 512 | 4096 |
> > > | Lumina-mGPT-1024 | 1/6 | 682 | 4096 |
> > >
> > > >**"Q2: Do the x-axis percentages denote KV retention ratio (kept tokens / original tokens) or pruning ratio (removed tokens / original tokens)?"**
> > >
> > > A2: **The x-axis percentages denote KV retention ratio.**
> > >
> > > For example, when Lumina-mGPT generates the 200th token, x = 10% means retaining only 20 image tokens in the KV cache, where the retained image KV cache consists of 10 prefix_tokens and 10 recent_tokens.
> > > **We will take the advice that we add a one-line definition in the caption, thank you very much.**
> > >
> > > >**"Q3: Is this auxiliary model the same architecture/checkpoint as the generator? How sensitive are the conclusions to this choice?"**
> > >
> > > A3: **The auxiliary model is the same model (same architecture and checkpoint) as the generator.**
> > >
> > > **First,** since each model processes images at almost completely different resolutions, using models with different architectures will lead to inconsistent entropy-indicated regions/token indices, resulting in inaccurate results. In general, this conclusion is very sensitive to the choice of model.
> > >
> > > **Second,** the experiments in Fig. 3(b) are mainly to explore the relationship between different entropy regions and compression ratios, therefore we used the same model to achieve the most accurate test.
> > >
> > > >**"Q4: Can the authors verify this claim and either cite related LM work or revise the statement and clarify the novelty as a vision-specific adaptation?"**
> > >
> > > A4: Sorry for the confusion. We state "There is currently no uncertainty-/entropy-aware KV compression method for LMs," which actually means that we claim our method is not a **direct transfer,** highlighting the novelty.
> > > Existing KV compression methods are mostly **fixed patterns or query-based,** such as H2O [1], SnapKV [2], and Quest [3].
> > > Applying these methods directly to visual AR generation models would lead to significant degradation in generation quality.
> > >
> > > **Our main contribution is proposing a KV sparsification method tailored for vision AR generation models.**
> > >
> > > Regarding entropy/uncertainty research, recent works have utilized entropy to help **reinforcement learning [4], byte-level modeling [5] and KV compression for LMs [6].**
> > > Uncomp [6] proposes using **Matrix Entropy within LMs** to analyze the information concentration of text sequences, thereby guiding KV compression. However, the aforementioned methods differ **significantly** from ours in terms of implementation and **cannot be simply** adapted or applied directly to our task.
> > > We illustrate the **differences** from SparseAR in two points:
> > >
> > > (i) Since LM addresses very large input sequences, **e.g. 30K,** primarily occurring in the **prefill stage,** while the input of visual AR generation models is **comparatively small** (text prompt has only about 30~100 tokens), the large prefill situation does not exist. Therefore, it is **difficult** to directly apply these prefill-dominated method to visual AR models.
> > >
> > > (ii) SparseAR proposes using **logit entropy** to guide KV compression. This intuition stems from our finding that this entropy value can effectively indicate the information density in an image. This is **vision-specific and absent in LMs,** which improve our novelty.
> > >
> > > All we want to claim is that SparseAR is neither a **direct transfer** nor a **simple adaptation** of existing methods. Rather, `SparseAR is specifically designed based on the insight that compression within low-entropy regions has a negligible impact on quality.`
> > >
> > >
> > > **References:**
> > >
> > > [1] [H2O.](https://arxiv.org/abs/2306.14048) NeurIPS 2023
> > >
> > > [2] [SnapKV.](https://arxiv.org/abs/2404.14469) NeurIPS 2024
> > >
> > > [3] [Quest.](https://arxiv.org/abs/2406.10774v1) ICML 2024
> > >
> > > [4] [Beyond the 80/20 Rule.](https://arxiv.org/abs/2506.01939) NeurIPS 2025
> > >
> > > [5] [BLT.](https://arxiv.org/abs/2412.09871) ACL 2025
> > >
> > > [6] [Uncomp.](https://arxiv.org/abs/2410.03090) EMNLP 2025

---

### Official Review · Reviewer_GDPc · 2026-03-09

**Soundness:** 3
**Presentation:** 3
**Significance:** 3
**Originality:** 2
**Overall Recommendation:** 4
**Confidence:** 5

**Summary:**

This paper addresses the inference efficiency bottleneck of autoregressive (AR) image generation models, where decoding long sequences of visual tokens incurs substantial KV cache memory and latency overhead. The authors argue that existing LLM-oriented KV compression methods are suboptimal for visual generation, as visual tokens are spatially redundant with highly non-uniform information density. To address this, they propose SparseAR, a training-free entropy-aware sparse attention method that uses the predictive entropy of the model as a proxy for local information density. The key insight is that high-entropy regions such as faces and fine textures require broader attention, while low-entropy regions such as flat backgrounds can tolerate aggressive KV sparsification. Based on this, SparseAR dynamically classifies each generation step into three entropy levels and allocates KV cache budgets accordingly. To avoid per-step entropy recomputation, the method exploits the spatial smoothness of entropy across adjacent positions and estimates current entropy from cached historical values. SparseAR is plug-and-play and is validated across four AR models on both text-to-image generation and image editing tasks, achieving substantial throughput improvements with largely preserved generation quality compared to prior KV compression baselines.

**Compliance With Llm Reviewing Policy:**

Affirmed.

**Final Justification:**

During rebuttal, the authors have addressed my concerns. Regarding WK1, the authors have clarified that SparseAR is not a straightforward transfer from existing LLM methods. In particular, the discovery of a new attention pattern in visual autoregressive models where generation positions are strongly correlated with corresponding positions in the conditional token sequence, which represents a meaningful and original contribution beyond mere adaptation. Regarding WK2, I appreciate the authors' acknowledgment of the presentation issues and their commitment to correcting them in the final version. Regarding WK3, the authors provide visualizations demonstrating that entropy reliably reflects information density in VAR and Meissonic, and offer a concrete discussion of how the method can be adapted to these alternative paradigms. In light of the above, I am updating my score to reflect the clarifications provided.

**Key Questions For Authors:**

Please see the weaknesses.

**Limitations:**

Yes.

**Strengths And Weaknesses:**

**Strengths**:

- The paper identifies a well-motivated and practically important problem. As AR image generation scales to higher resolutions, KV cache overhead becomes an increasingly critical bottleneck, and the observation that visual tokens have fundamentally different redundancy patterns from text tokens is well-supported by the preliminary analysis in Figure 2 and Figure 3.
- The use of predictive entropy as a lightweight proxy for information density is intuitive and grounded in empirical evidence. The authors carefully validate the entropy-compression relationship before building the full method, lending credibility to the core design choice.
- SparseAR is training-free and plug-and-play, which lowers the barrier to adoption and makes the method broadly applicable across different AR model families without architectural modifications.
- The method is evaluated on both text-to-image generation and image editing tasks across multiple models and resolutions, providing a reasonably comprehensive empirical picture.

**Weaknesses**:

- The technical novelty is limited. Using uncertainty or entropy to guide adaptive KV compression has been explored in the LLM literature, and the primary contribution of this paper is transferring that idea to visual AR generation. The adaptation, while reasonable, does not introduce fundamentally new algorithmic insights.
- The paper requires careful proofreading before publication. Specifically, Figure 5 contains an unfilled placeholder reporting "x% inference speedup", the throughput numbers reported in Table 1 are inconsistent with the speedup figures claimed in the main text, and there is a citation formatting issue at line 780. These issues should all be corrected before publication.
- The method is only validated on token-by-token autoregressive models such as LlamaGen and Lumina-mGPT. It remains unclear whether SparseAR would transfer effectively to other popular AR paradigms such as MAR and VAR, which generate tokens in fundamentally different orders and structures. It would strengthen the paper considerably to include experiments or at least a discussion of how the entropy-based locality assumption holds under these alternative generation frameworks.

---

> ### Author Rebuttal · Authors · 2026-03-29
>
> We appreciate your acknowledgment that **"The paper identifies a well-motivated and practically important problem."** Also, we appreciate that you think the method is **"training-free and plug-and-play, which lowers the barrier to adoption".**
>
> >**"WK1: The technical novelty is limited. Using uncertainty or entropy to guide adaptive KV compression has been explored in the LLM literature, and the primary contribution of this paper is transferring that idea to visual AR generation. The adaptation, while reasonable, does not introduce fundamentally new algorithmic insights."**
>
> Thank you for your question. We will now describe our method from several aspects to explain why it is not a direct transfer from existing LLM methods and to highlight our unique contributions.
>
> * **"Relationship with Existing KV Compression Methods for LLMs"**:
>
>     The motivation for this work stems from the discovery that KV Compression is **highly effective and well-researched** in Large Language Model, while there is virtually no such work on visual AR models, especially for tasks like image generation and image editing. Furthurmore, there is currently no work on "using uncertainty or entropy to guide adaptive KV compression.
>
> * **"Core Contribution in the Visual Autoregressive Generation Models"**:
>
>      (i) ***A Vision-Specific KV Sparsification Method Unique to Visual AR Models.*** As mentioned in Figure 1 and Table 3 of the main text, we directly attempted to apply classic LLM KV compression methods for text-to-image generation and image editing tasks. However, when attempting to **directly transfer** these methods to visual autoregressive models, the **inherent difference** between visual and text tokens can cause erroneous results in highly complex regions, affecting all subsequent generation. This paper addresses this issue by proposing a KV Sparsification method unique to visual AR models, demonstrating excellent performance.
>
>      (ii) ***Novel Efficient Sparse Attention for Image Editing.*** AR image generation models such as Blip3o-Next [1] and Lumina-mGPT2.0 [2] have **demonstrated powerful image editing capabilities,** thus indicating the potential of the autoregressive image generation approach. We are **the first** to discover a common attention pattern exhibited by these models in image editing tasks: the generation position is strongly correlated with the corresponding position in the conditional token sequence. Therefore, based on this characteristic, we further improve our **entropy-aware KV Sparsification** method and apply it to image editing tasks.
>
> >**"WK2: The paper requires careful proofreading before publication. Specifically, Figure 5 contains an unfilled placeholder reporting "x% inference speedup", the throughput numbers reported in Table 1 are inconsistent with the speedup figures claimed in the main text, and there is a citation formatting issue at line 780."**
>
> Thank you for your meticulous review and suggestions. Specifically, the speedup ratios presented in Figure 5 and the main text are derived from direct estimates based on our measured throughput results across various configurations. In the main text, we employed phrases such as **"at least"** and **"nearly"** to qualify these figures. We sincerely apologize for any lack of clarity about the precise numerical values ​​of the speedup ratios and will endeavor to rectify this in future revisions if possible.
>
> >**"WK3: The method is only validated on token-by-token autoregressive models such as LlamaGen and Lumina-mGPT. It remains unclear whether SparseAR would transfer effectively to other popular AR paradigms such as MAR and VAR, which generate tokens in fundamentally different orders and structures."**
>
> **SparseAR can be transferred to other generative architectures with only minor modifications.** To begin, we demonstrate that the VAR [3] and Meissonic [4] can also indicate information density based on entropy. Our visualization results are shown in **[Anonymous Github Link](https://anonymous.4open.science/r/tt_tt_374698/.)**. Next, we will explain why our method is transferable. For VAR, it generates images scale-by-scale from low resolution. All tokens at the current scale share the cached KV from previous scales, so entropy can be estimated by mapping the entropy from the previous scale to the current scale.  For Meissonic-like model, this approach uses bidirectional attention and generates tokens at completely random locations, prioritizing low-entropy locations. Therefore, we envision directly using sparse attention after several generation steps, as the model still expects to receive low-entropy tokens, thus allowing for attention computation.
>
>
>
> **References:**
>
> [1] [Blip3o-Next.](https://arxiv.org/abs/2510.15857)
>
> [2] [Lumina-mGPT 2.0.](https://arxiv.org/abs/2507.17801)
>
> [3] [VAR](https://arxiv.org/abs/2404.02905).
>
> [4] [Meissonic](https://arxiv.org/abs/2309.17453).

---

> > ### Author Rebuttal · Reviewer_GDPc · 2026-04-03
> >
> > Thank the authors for their thorough rebuttal. My main concerns are fully resolved. I have raised my score.

---

> > > ### Author Response · Authors · 2026-04-04
> > >
> > > **We sincerely appreciate your detailed feedback and the time you have taken to share your thoughts.** It is encouraging to know that our responses have been helpful, and we remain fully committed to refining our paper based on your insights.
> > > Furthermore, we greatly appreciate your **upgrade from "3" to "4"** and thank you very much for recognizing our rebuttal.
> > >
> > > **We look forward to your continued comments and thank you once again for your careful review.**

---

### Official Review · Reviewer_EFVL · 2026-03-13

**Soundness:** 4
**Presentation:** 3
**Significance:** 3
**Originality:** 3
**Overall Recommendation:** 4
**Confidence:** 4

**Summary:**

The paper proposes SparseAR, a training-free, entropy-aware dynamic KV cache sparsification method tailored for autoregressive (AR) image generation and editing. The authors compellingly observe that unlike text, visual tokens exhibit highly non-uniform spatial information density. To exploit this, SparseAR leverages predictive entropy as a signal: it retains the full KV cache for high-entropy (complex, information-rich) regions while aggressively compressing the cache for low-entropy (flat, redundant) regions. Evaluated on multiple AR models (e.g., LlamaGen, Lumina-mGPT, EditAR), the proposed method demonstrates substantial improvements in inference throughput and latency without compromising—and occasionally improving—generation quality (FID/CLIP scores).

**Compliance With Llm Reviewing Policy:**

Affirmed.

**Final Justification:**

I appreciate the authors' rebuttal and will maintain my positive rating.

**Key Questions For Authors:**

(1) Latency Breakdown: Could the authors provide a granular wall-clock latency breakdown? Specifically, what is the exact time overhead incurred by the dynamic entropy estimation, spatial-temporal neighbor mapping, and KV cache reallocation at each step, compared to the time saved in the actual Attention computation?

(2) Worst-Case Efficiency Bounds: How does SparseAR perform in extreme edge cases where the entire image consists of highly complex textures (e.g., crowds, dense foliage)? A quantitative analysis of the speedup/throughput degradation in these high-entropy scenarios would strengthen the paper.

(3) Adaptability to Multi-Scale AR: How would the proposed spatial-temporal neighbor heuristic and entropy thresholding adapt to next-scale AR models (like VAR) where tokens are generated resolution-by-resolution rather than row-by-row?

**Limitations:**

The authors have honestly acknowledged the limited efficiency gains in globally high-entropy or low-entropy scenarios. However, they should also explicitly discuss the computational overhead of the dynamic routing and entropy calculation itself, especially as VQ codebook sizes continue to scale up in future AR models.

**Strengths And Weaknesses:**

Strengths:

(1) Insightful and Domain-Specific Motivation: The observation that visual token redundancy is spatially non-uniform is highly insightful. By moving beyond fixed-pattern or modality-agnostic KV compression methods (such as H2O or StreamingLLM), the authors provide a principled, vision-specific solution.

(2) Training-Free and Plug-and-Play: The method elegantly estimates local entropy based on historical caches and spatial-temporal neighbors, eliminating the need for external draft models or costly fine-tuning. This makes it highly versatile across different AR architectures.

(3) Strong Empirical Results: The experimental validation is solid. The trade-off between FLOPs/latency and generation quality (FID/CLIP) is convincing, particularly the finding that pruning redundant distant tokens can sometimes reduce noise and improve text alignment.

Weaknesses:

(1) Overhead of Entropy Calculation: While SparseAR reduces the length of the KV cache for attention computation, calculating the predictive entropy itself (Eq. 1) requires summing over the entire VQ codebook (which can range from 16K to 100K+ entries). The paper lacks a detailed latency breakdown indicating how much computational overhead and CUDA kernel synchronization delay this dynamic estimation and allocation introduces.

(2) Degradation in Extreme Scenarios: As briefly mentioned in the limitations, the method may struggle with globally high-entropy images (e.g., a complex Persian carpet or a dense starry sky). In such worst-case scenarios, if the framework defaults to retaining the full KV cache across the board, the efficiency gains might completely vanish. The empirical bounds of this degradation are not thoroughly evaluated.

(3) Compatibility with Next-Scale AR Paradigms: The current spatial-temporal neighbor construction (Eq. 2 and 3) assumes a traditional raster-scan decoding order. However, the field is rapidly shifting towards next-scale or multi-scale AR models (e.g., VAR, MAR), where the token sequence represents progressive resolutions rather than purely spatial scanning. The adaptability of SparseAR to these emerging paradigms remains unaddressed.

---

> ### Author Rebuttal · Authors · 2026-03-28
>
> Thank you for your thorough and meticulous review. We appreciate your noting that **"The observation that visual token redundancy is spatially non-uniform is highly insightful."**
>
> >**"Wk1: Could the authors provide a granular wall-clock latency breakdown?"**
>
> * **The latency incurred by entropy calculation is almost negligible.** Calculating the predictive entropy itself (Eq. 1) requires summing over the entire VQ codebook. However, the forward process of the AR model has **already implemented softmax** calculation of the context, and obtaining entropy only requires summing the softmax results, including **thousands of floating-point operations,** which can be completed in just **a few microseconds.**
>
> To provide further clarity, We demonstrate this using the following pseudocode.
>
>     # Entropy computation
>
>     # ── AR Forward Propagation ───
>     context = tokens[:i + 1]  # first i+1 tokens
>     logits  = AR_model(context)
>     probs   = softmax(logits)
>
>     # Shannon Entropy (Eq. 1)
>     # probs[k] is the probability of the k-th codebook entry
>     # entropy_i is calculated via microsecond-level summation operations.
>     entropy_i = -Σ_{k=1}^{V}  probs[k] · log(probs[k])
>
> * **Dynamic entropy estimation requires only a simple O(1) complexity operation.** Following each generation step, we compute the entropy at the current position and **store its value in a dictionary**, indexed by its corresponding position. When the entropy value is needed to guide the estimation, only **microsecond-level** dictionary lookup operations are required.
> * **KV cache reallocation incurs minimal latency overhead.** We obtain the compressed KV representation by applying slicing and indexing operations to the vectors within the KV Cache. This is a purely memory-indexing operation that incurs virtually **no computational overhead.**
> To clearly demonstrate the latency overhead of each operation, we conducted comparative experiments on test sets containing 1,000 and 5,000 samples.
>
> The results are presented in the table below.
>
> | SparseAR | Lumina-mGPT-512 (Text-to-Image) | Blip3o-Next (Image-Editing) |
> |---|:---:|:---:|
> | **KV Reallocation**  |0.134s (1K) / 1.779s (5K) | 0.289s (1K) / 3.487s (5K) |
> | **Attention Computation** | 1.51s (1K) / 5.504s (5K) |  2.24s (1K) / 8.331s (5K) |
> | **Query-Based Attention** |1.02s (1K) / 4.488s (5K) | 1.882s (1K) / 6.905s (5K) |
>
> `The results show that our methods require almost no latency overhead compared to attention computation.`
>
> >**"WK2: How does SparseAR perform in extreme edge cases where the entire image consists of highly complex textures?"**
>
> **SparseAR maintains a certain speedup even in extreme scenarios, especially for image editing.** We present two sets of experimental results: (i) The percentage and the specific number of tokens for each entropy category when generating 5000 samples; (ii) The quality and efficiency on a test set containing 800 samples, each image containing 80% high entropy.
> The results are shown in the table below, where L represents Lumina-mGPT-512 and B represents Blip3o-Next.
>
> |Models|Low entropy|Mid entropy|High entropy|
> |:---:|:---:|:---:|:---:|
> |Base-L|29.3% (2,999,719) | 30.0% (3,071,385) | 40.6% (4,156,608) |
> |Base-B|32.8% (3,674,983) | 25.9% (4,163,277) | 48.2% (6,511,329) |
>
> | Model | FID | FLOPS |
> |:---:|:---:|:---:|
> | Base-L | 25.67 | 7.63 |
> | SparseAR-L | 26.11 | 7.35 |
> | Base-B | 11.17 | 29.77 |
> | SparseAR-B | 10.56 | 25.11 |
>
> In the overall sample, low-entropy tokens are only about 10% fewer than high-entropy ones. And SparseAR achieves ~19% improvement in extreme scenarios.
> `These results demonstrate that SparseAR can achieve significant speedup in extreme scenarios while remaining effective across diverse cases.`
>
> >**"WK3: How would the proposed spatial-temporal neighbor heuristic and entropy thresholding adapt to next-scale AR models (like VAR)?"**
>
> The proposed spatial-temporal neighbor heuristic **can be adapted** to next-scale AR [1] models with slight modifications. First, the entropy value of the next-scale model can also reflect the information density of the generated region, as demonstrated in **[Anonymous Github Link](https://anonymous.4open.science/r/tt_tt_374698/.)**. Next-scale model generates images scale-by-scale from low resolution. All tokens at the current scale share the cached KV from previous scales, so entropy can be estimated by mapping the entropy from the previous scale to the current scale. We demonstrate this using the following pseudocode:
>
>     # Scale s-1 → Scale s
>     tokens_prev  = r_{s-1}
>     pr_prev  = AR_model_softmax(tokens_prev)
>     entro_prev = - (pr_prev * log(pr_prev + ε)).sum(dim=1)
>
>     # Spatial Mapping
>     h_s, w_s = scales[s]
>
>     i_idx = (arange(h_s)[:, None] // 2).expand(h_s, w_s)  # i // 2
>     j_idx = (arange(w_s)[None, :] // 2).expand(h_s, w_s) # j // 2
>     entro_est = entro_prev[:, i_idx, j_idx]
>
> **References:**
>
> [1] [VAR](https://arxiv.org/abs/2404.02905). NeurIPS 2024

---

> > ### Author Rebuttal · Reviewer_EFVL · 2026-04-02
> >
> > Thank you for the rebuttal; my concerns are addressed and I maintain my positive score.

---

> > > ### Author Response · Authors · 2026-04-02
> > >
> > > **Thank you very much for your reply and for maintaining the positive score.** We truly appreciate your engagement in the discussion and are pleased that our previous response addressed your concerns.
> > > We remain eager to learn from your feedback and improve the paper further.
> > >
> > > **Thank you again for your time and thoughtful consideration.**

---

### Decision · Program_Chairs · 2026-04-30

**Decision:**

Accept (regular)

**Comment:**

This paper addresses the memory footprint and latency bottlenecks inherent in Autoregressive (AR) image generation by introducing SparseAR. SparseAR is a training-free, entropy-aware dynamic KV cache sparsification method. The reviewers appreciated the motivation behind the paper. Adapting KV cache compression to AR visual generation is a highly relevant, timely, and underexplored research direction. The proposed training-free paradigm is appreciated for its potential scalability and practical utility. Therefore, the AC recommends acceptance.